# Loss of FLCN-FNIP1/2 induces a non-canonical interferon response in human renal tubular epithelial cells

Iris E Glykofridis[1]*, Jaco C Knol[2], Jesper A Balk[1], Denise Westland[3], Thang V Pham[2], Sander R Piersma[2], Sinéad M Lougheed[2], Sepide Derakhshan[4], Puck Veen[1], Martin A Rooimans[1], Saskia E van Mil[1], Franziska Böttger[2], Pino J Poddighe[5], Irma van de Beek[5], Jarno Drost[4], Fried JT Zwartkruis[3], Renee X de Menezes[6], Hanne EJ Meijers-Heijboer[1], Arjan C Houweling[5], Connie R Jimenez[2]*, Rob MF Wolthuis[1]*

[1]Amsterdam UMC, location VUmc, Vrije Universiteit Amsterdam, Clinical Genetics, Cancer Center Amsterdam, Amsterdam, Netherlands; [2]Amsterdam UMC, location VUmc, Vrije Universiteit Amsterdam, Medical Oncology, Cancer Center Amsterdam, Amsterdam, Netherlands; [3]University Medical Center Utrecht, Center for Molecular Medicine, Molecular Cancer Research, Universiteitsweg, Utrecht, Netherlands; [4]Princess Máxima Center for Pediatric Oncology, Oncode Institute, Heidelberglaan, Utrecht, Netherlands; [5]Amsterdam UMC, location VUmc, Vrije Universiteit Amsterdam, Clinical Genetics, Amsterdam, Netherlands; [6]NKI-AvL, Biostatistics Unit, Amsterdam, Netherlands

*For correspondence:
i.glykofridis@amsterdamumc.nl (IEG);
c.jimenez@amsterdamumc.nl (CRJ);
r.wolthuis@amsterdamumc.nl (RMFW)

Competing interests: The authors declare that no competing interests exist.

**Abstract** Germline mutations in the Folliculin (*FLCN*) tumor suppressor gene cause Birt–Hogg–Dubé (BHD) syndrome, a rare autosomal dominant disorder predisposing carriers to kidney tumors. *FLCN* is a conserved, essential gene linked to diverse cellular processes but the mechanism by which *FLCN* prevents kidney cancer remains unknown. Here, we show that disrupting *FLCN* in human renal tubular epithelial cells (RPTEC/TERT1) activates TFE3, upregulating expression of its E-box targets, including RRAGD and GPNMB, without modifying mTORC1 activity. Surprisingly, the absence of FLCN or its binding partners FNIP1/FNIP2 induces interferon response genes independently of interferon. Mechanistically, FLCN loss promotes STAT2 recruitment to chromatin and slows cellular proliferation. Our integrated analysis identifies STAT1/2 signaling as a novel target of FLCN in renal cells and BHD tumors. STAT1/2 activation appears to counterbalance TFE3-directed hyper-proliferation and may influence immune responses. These findings shed light on unique roles of FLCN in human renal tumorigenesis and pinpoint candidate prognostic biomarkers.

## Introduction

Renal cell carcinoma (RCC) is the most common form of kidney cancer representing up to 5% of newly identified cancer cases (*Ferlay et al., 2019*; *Lopez-Beltran et al., 2006*; *Siegel et al., 2018*). Generally, RCCs are diagnosed in adults, with the exception of translocation RCC, which is driven by a hyper-activated fusion protein of the transcriptional activators TFE3 or TFEB and comprises 20–75% of RCCs in childhood (*Ambalavanan and Geller, 2019*; *Caliò et al., 2019*; *Lee et al., 2018*). Birt-Hogg-Dubé syndrome (BHD) is a dominantly inherited kidney cancer syndrome caused by mono-allelic germline loss-of-function mutations of the essential and conserved Folliculin (*FLCN*) gene (*Nahorski et al., 2011*; *Nickerson et al., 2002*). The lifetime risk for BHD patients to develop RCC is ~10 times higher than for the unaffected population (*Houweling et al., 2011*;

*Pavlovich et al., 2005*; *Toro et al., 2008*; *Zbar et al., 2002*). BHD patients are predisposed to bilateral and multifocal renal tumors (*Nickerson et al., 2002*; *Schmidt et al., 2005*; *Zbar et al., 2002*) and dependent on surveillance by renal imaging for early detection and curative treatment prior to metastasis (*Johannesma et al., 2019*). Loss of heterozygosity, by gene silencing or an inactivating somatic mutation of the wild-type *FLCN* allele, is a prerequisite for kidney cancer development in BHD patients (*Vocke et al., 2005*).

Much of our understanding of BHD-related RCC is based on studies in BHD animal models (*Chen et al., 2008*; *Chen et al., 2015*; *Hudon et al., 2010*) and a BHD kidney tumor-derived cell line (*Yang et al., 2008*). These studies have connected *FLCN* to diverse cellular processes including mitochondrial biogenesis, stress resistance, autophagy, membrane trafficking, stem cell pluripotency, and ciliogenesis (*Baba et al., 2006*; *Betschinger et al., 2013*; *Dunlop et al., 2014*; *Hasumi et al., 2012*; *Laviolette et al., 2013*; *Luijten et al., 2013*; *Nookala et al., 2012*; *Possik et al., 2014*). The FLCN protein has been reported to affect multiple regulatory factors including mTOR, AMPK, HIF1, TGF-β, and Wnt (*Baba et al., 2006*; *De Zan et al., 2020*; *El-Houjeiri et al., 2019*; *Hong et al., 2010b*; *Khabibullin et al., 2014*; *Mathieu et al., 2019*; *Preston et al., 2011*; *Yan et al., 2016*). Nevertheless, the mechanism by which FLCN loss induces tumorigenesis is largely unknown. Conflicting results, such as activating or inhibitory effects of FLCN on mTOR signaling (*Baba et al., 2006*; *Bastola et al., 2013*; *Hartman et al., 2009*; *Hasumi et al., 2009*; *Hudon et al., 2010*; *Napolitano et al., 2020*; *Takagi et al., 2008*; *Tsun et al., 2013*), and the range of the processes attributed to FLCN loss, prohibit a clear understanding of the pathways by which FLCN suppresses renal tumorigenesis.

Here, we present the molecular and cellular consequences of knocking out *FLCN* or its binding partners *FNIP1/FNIP2* in a human renal proximal tubular epithelial cell model, representing the cells of origin of RCC (*Holthöfer et al., 1983*). We performed RNA sequencing (RNAseq) and proteomics, followed by pathway analyses and mining of regulatory promotor motifs of differentially expressed genes, revealing that FLCN loss induces two separate transcriptional signatures. The first is characterized by E-box controlled genes and confirms TFE3 as a main target of the FLCN-FNIP1/2 axis (*El-Houjeiri et al., 2019*; *Endoh et al., 2020*; *Hong et al., 2010b*; *Petit et al., 2013*). Secondly, we discovered that loss of FLCN-FNIP1/2 induces a set of genes under control of interferon-stimulated response elements (ISREs). The ISRE gene activation program is directed by upregulate STAT1 and STAT2 and may explain why loss of the FLCN tumor suppressor, paradoxically, reduces cellular proliferation. We propose that TFE3 and STAT1/2 are the two main, independent transcriptional effectors of FLCN-FNIP1/2 loss in human renal epithelial cells. Preliminary data indicate that these gene networks may also be activated in BHD tumors. Taken together, our findings may help the development of prognostic biomarkers or targeted therapies.

## Results

### Knocking out FLCN activates TFE3 in renal proximal tubular cells

To study the effects of FLCN loss in a context relevant for oncogenesis, we used an immortalized, diploid renal proximal tubular epithelial cell line (RPTEC/TERT1, ATCC CRL-4031; *Wieser et al., 2008*, hereafter called RPTEC) as a model system, which retains the capacity to form 3D tubular structures (*Figure 1A*). First, we constructed an inducible Cas9-expressing RPTEC cell line and verified doxycycline-induced Cas9 protein expression by immunoblots (*Figure 1B*). Because a TP53-dependent DNA damage response prohibits effective gene editing in some cell types (*Haapaniemi et al., 2018*; *Ihry et al., 2018*), we simultaneously knocked-out *TP53* and *FLCN* to improve targeting efficiency. Guide RNAs (gRNAs) targeting 5' coding exons of *TP53* and *FLCN* (*Figure 1C*) were co-transfected and targeted exons of single-cell derived clones were sequenced. *Indel* analysis showed that each clone carried a unique genetic disruption of *FLCN* and *TP53* (*Figure 1—figure supplement 1A*). By karyotypic analysis, we found that four out of six knock out (KO) cell lines had become aneuploid (*Figure 1D*). We conclude, however, that aneuploidy occurred spontaneously during cell line formation because it appeared independently of FLCN, TP53, or inducible Cas9 status (*Figure 1D*). At a later stage, we indeed developed diploid *FLCN* knock out RPTEC cell lines using more advanced gRNA delivery assays (see below). However, considering that kidney tumors are often aneuploid (*Kardas et al., 2005*; *Morlote et al., 2019*), we first used the cell

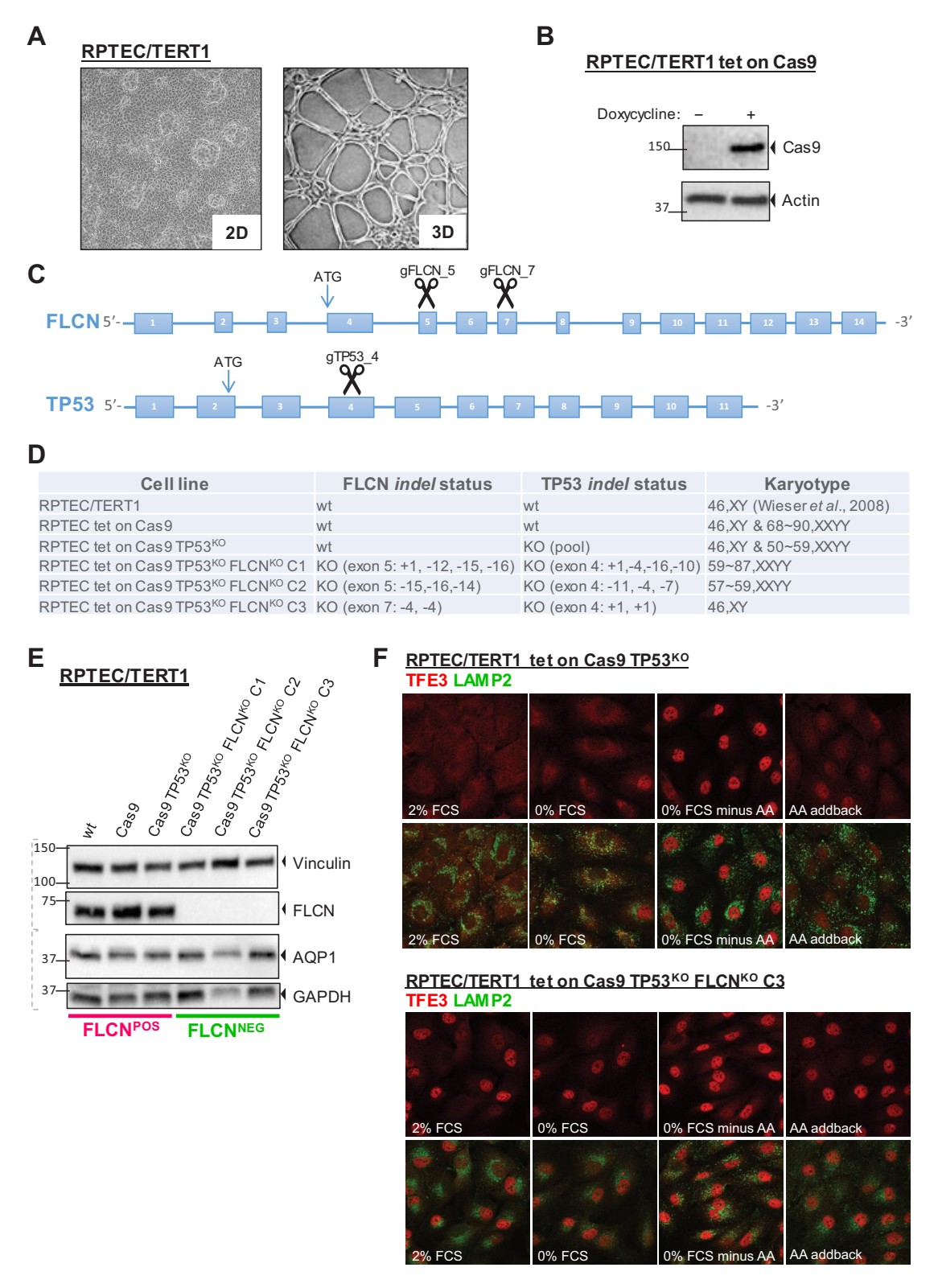

**Figure 1.** Renal proximal tubular epithelial cells as a model for FLCN loss. (**A**) Brightfield images (×50 magnification) of a human renal proximal tubular epithelial cell model (RPTEC/TERT1). Left image shows 2D culture of cells with typical dome formation. Right image shows 3D tubular structures that form when RPTECs are cultured according to *Secker, 2018*. (**B**) Doxycycline-inducible Cas9 expression of RPTEC tet-on Cas9 cell line. Cas9 protein expression after 24 hr treatment with 10 ng/ml doxycycline was assessed by immunoblotting. Experiment was performed twice. (**C**) CRISPR/Cas9-

*Figure 1 continued on next page*

Figure 1 continued

mediated knockout strategy of FLCN and TP53 in RPTEC/TERT1 cells; gRNAs were designed to target early exons of FLCN and TP53 coding regions. (D) Overview of *FLCN* and *TP53 indel* status and karyotype per selected cell line clone. Cell-line-specific Sanger sequence chromatograms are shown in *Figure 1—figure supplement 1A*. (E) Western blot of FLCN protein levels of indicated cell line clones. Expression of renal proximal tubular-specific marker AQP1 is shown as a control. Dotted lines indicate separate blots. Western blot of TP53 protein levels is shown in *Figure 1—figure supplement 1B*. (F) Immunofluorescence staining of TFE3 and lysosomal marker LAMP2 show enhanced nuclear TFE3 upon FLCN loss independent of nutrient availability. FCS = fetal calf serum, AA = amino acids. Staining of FLCN[KO] RPTEC C1 and C2 are shown in *Figure 1—figure supplement 1C*.

The online version of this article includes the following figure supplement(s) for figure 1:

**Figure supplement 1.** Characterization of TP53[KO] and FLCN[KO] cell lines.

lines of *Figure 1D* to identify FLCN-specific effects that occur independently of karyotype. We started by comparing two groups of cell lines: RPTEC ('WT'), RPTEC tet-on Cas9 ('Cas9'), and RPTEC tet-on Cas9 TP53[-/-] ('TP53[KO]') were assigned to the FLCN[POS] group, while three individually isolated RPTEC tet-on Cas9 *TP53[-/-] FLCN[-/-]* clones ('FLCN[KO] C1','FLCN[KO] C2', and 'FLCN[KO] C3') were assigned to the FLCN[NEG] group. C1 and C2 were created by gRNAs targeting FLCN exon 5 ('gFLCN_5') and C3 by gRNAs targeting *FLCN* exon 7 ('gFLCN_7'). Loss of FLCN and TP53 protein expression was confirmed by immunoblots, which also showed that expression of the renal proximal tubular marker aquaporin-1 (AQP1) was unchanged (*Bedford et al., 2003*; *Figure 1E* and *Figure 1—figure supplement 1B*).

Previous studies reported that FLCN prevents nuclear localization of TFE3 under nutrient-rich conditions. TFE3, and its close family member TFEB, are transcription factors directing an autophagy and stress tolerance gene program under growth restrictive conditions (*Hong et al., 2010a*; *Wada et al., 2016*). To investigate the status of this pathway in our *FLCN* knock-out RPTEC cell models, we visualized TFE3 localization in the presence or absence of FLCN. Immunofluorescence co-staining of TFE3 and the lysosomal marker LAMP2 in either fed, starved, or amino acid (AA)-depleted conditions are shown in *Figure 1F* and *Figure 1—figure supplement 1C*. RPTEC cells showed cytoplasmic TFE3 under normal growth conditions but TFE3 completely translocated to the nucleus after withdrawal of serum and amino acids (*Figure 1F*, upper panels). In contrast, TFE3 localized in the nucleus of the majority of cells from each of the three RPTEC FLCN[NEG] cell lines, independent of starvation or refeeding, revealing that FLCN[NEG] cells fail to convey proper nutrient sensing and TFE3 responses (*Figure 1F*, lower panels and *Figure 1—figure supplement 1C*). We conclude that, in the absence of FLCN, TFE3 is constitutively active in renal epithelial cells, confirming TFE3 as a bona fide target of the FLCN tumor suppressor (*El-Houjeiri et al., 2019*; *Wada et al., 2016*).

## Overlapping transcriptomic and proteomic alterations induced by FLCN loss

Subsequently, we determined changes in gene transcription and protein expression patterns in our FLCN[POS] and FLCN[NEG] RPTEC cell lines by mRNA sequencing (RNAseq) and proteomic workflows shown in *Figure 2A*. For specificity, we only compared profiles of the FLCN[POS] and FLCN[NEG] cell lines between the groups defined by rectangles in *Figure 2—figure supplement 1A* (for RNAseq) and *Figure 2—figure supplement 2A* (for proteomics). To correct for possible clonal effects on global transcription, three different TP53 knock-out clones were included in the RNAseq analysis. We used edgeR (*Robinson et al., 2010*) to identify FLCN-related effects and visualized differential expressed genes in volcano plots (*Figure 2B*). Green circles show genes upregulated in FLCN[NEG] cells and pink circles indicate genes expressed at a higher level in control (FLCN[POS]) cells. The threshold line represents a false discovery rate (FDR) of <0.05. The edgeR results of RPTEC FLCN[POS] versus FLCN[NEG] comparison are in *Supplementary file 1* and a volcano plot with annotated gene names is shown in *Figure 3—figure supplement 1A*. Interestingly, the majority of significantly altered genes was induced, rather than repressed, by FLCN inactivation: in FLCN[NEG] cells 426 genes were upregulated at least tenfold while only 62 genes were downregulated to the same extent (FDR < 0.05, *Figure 2—figure supplement 1D*, lower panel). To explore downstream consequences of FLCN loss at the protein level, we used mass spectrometry-based proteomics, identifying 5755 different proteins (*Figure 2—figure supplement 2C and E*). Comparative analysis of FLCN[POS] versus

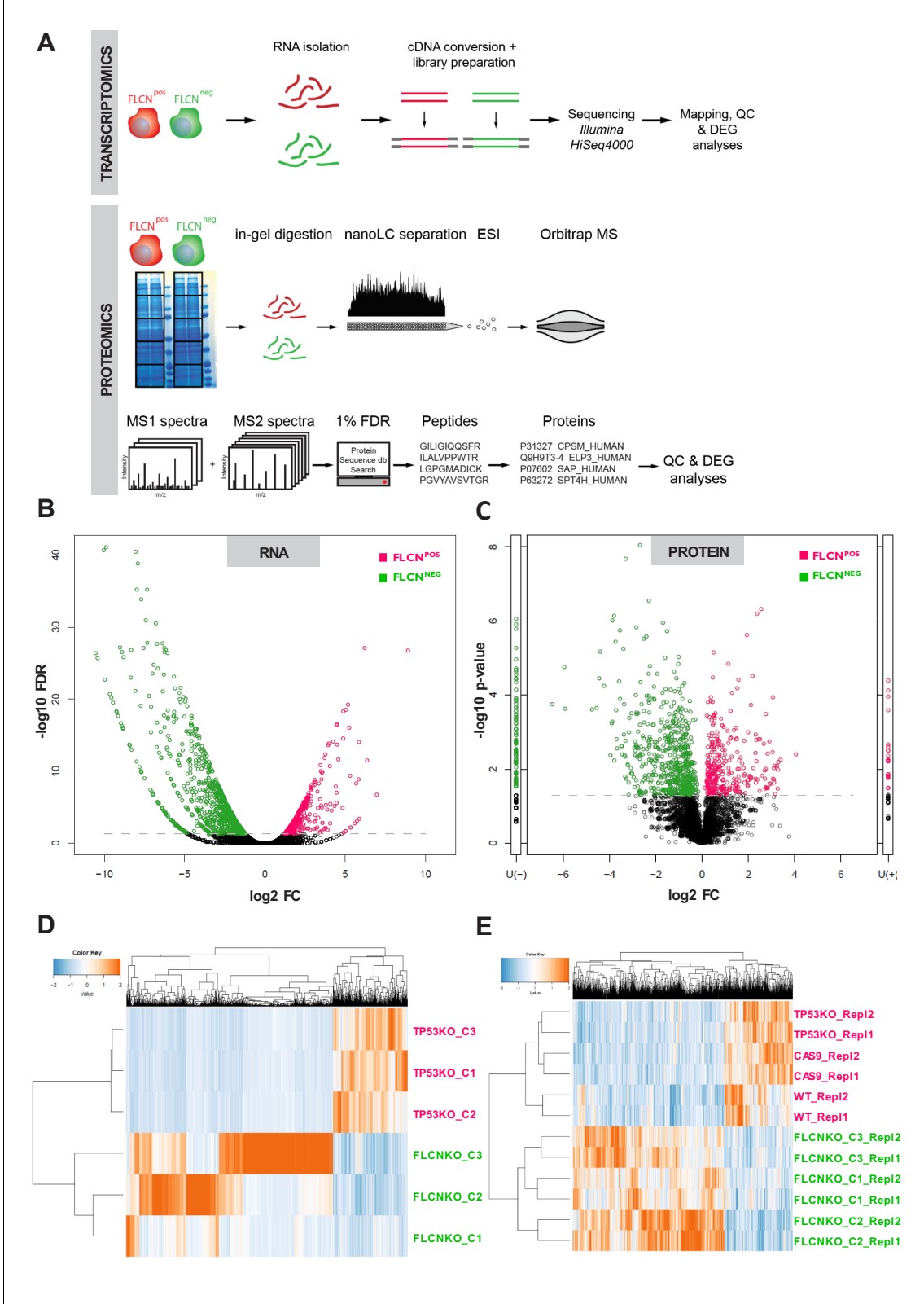

**Figure 2.** Integrated transcriptomic and proteomic analyses of renal tubular FLCN loss. (**A**) Schematic overview of transcriptomic and proteomic workflows. (**B**) Volcano plot showing significantly increased or decreased expression of genes in FLCN^POS vs. FLCN^NEG comparison derived from transcriptomic analysis. Colored circles above threshold line are FDR < 0.05; statistical details can be found in Materials and methods section. (**C**) Volcano plot showing (significantly) increased or decreased expression of proteins in FLCN^POS vs. FLCN^NEG comparison derived from proteomic

*Figure 2 continued on next page*

*Figure 2 continued*

analysis. Colored circles above threshold line are p<0.05. U(-) column shows proteins uniquely detected in FLCN<sup>NEG</sup>, U(+) column shows proteins uniquely detected in FLCN<sup>POS</sup>. Statistical details can be found in the Materials and methods section. (D) Hierarchical clustering based on FLCN-dependent differential mRNA expression. (E) Hierarchical clustering based on FLCN-dependent differential protein expression.

The online version of this article includes the following figure supplement(s) for figure 2:

**Figure supplement 1.** Comparative analyses of RPTEC FLCN<sup>POS</sup> vs. FLCN<sup>NEG</sup> cell line pairs.

**Figure supplement 2.** GeLC-MS/MS-based proteomics of RPTEC FLCN<sup>POS</sup> vs. FLCN<sup>NEG</sup> cell line pairs.

FLCN<sup>NEG</sup> cells using normalized spectral counts (beta-binomial test, [*Pham et al., 2010*]), identified the differential expression pattern presented in *Figure 2C* (threshold line indicates p-values of <0.05). The U(-), and U(+), columns next to the volcano plots show proteins that were detected in only one of the conditions. Volcano plots annotated with corresponding protein names in are shown in *Figure 3—figure supplement 1A* and the complete comparative analysis can be found in *Supplementary file 1*.

In concordance with our RNAseq data, FLCN loss predominantly induced protein expression: 209 proteins were expressed at least fivefold higher in FLCN<sup>NEG</sup> cells, versus 41 proteins that were similarly downregulated, out of a total of 914 differentially expressed proteins (p<0.05 and average count >1.5; *Figure 2—figure supplement 2E*, lower right). Conversely, as a control, deletion of the transcriptional activator TP53 alone (*Farmer et al., 1992*; *Fields and Jang, 1990*; *Vogelstein and Kinzler, 1992*) resulted in downregulation of gene expression (*Figure 2—figure supplement 1E*, right, volcano plots of differential expression analysis by edgeR). Expression of differential RNAs showed variation between FLCN<sup>NEG</sup> RPTEC clones (*Figure 2D*, *Figure 2—figure supplement 1C*). Nevertheless, hierarchical cluster analysis of differentially expressed proteins showed that replicates clustered together with a clear separation between the two groups (*Figure 2D,E*, *Figure 2—figure supplement 2D*), pointing to clonally independent effects of FLCN loss. Taken together, these results identify FLCN as a powerful repressor of gene expression.

## FLCN loss activates a specific set of TFE3 target genes in kidney epithelial cells

To determine which TFE3/TFEB (TFE) driven genes are activated by FLCN loss, we collected 248 previously reported TFE targets (*Hong et al., 2010a*; *Martina et al., 2014*; *Palmieri et al., 2011*; *Santaguida et al., 2015*) and performed *k-means* Pearson correlation clustering of our FLCN<sup>POS</sup> vs. FLCN<sup>NEG</sup> RPTEC RNAseq data to categorize the effects. Interestingly, we found a specific subset of 115 known TFE targets to be upregulated in all three FLCN<sup>NEG</sup> cell lines (*Figure 3A*, cluster 3, boxed yellow, *Supplementary file 2*). Consistent upregulation of TFE target genes FNIP2, GPNMB, SQSTM1, RRAGC, GABARAP, ARHGAP12, AMDHD, WIPI1 and the more recently identified tumor growth-promoting TFE target RRAGD (not included in *Figure 3A*; *Di Malta et al., 2017*) was validated in all three FLCN<sup>NEG</sup> RPTEC clones using quantitative RT-PCR (*Figure 3B*). Western blots of GPNMB, RRAGD, SQSTM1, and FNIP2, proteins associated with lysosome function, also showed these were strongly upregulated by FLCN inactivation, irrespective of TP53 status (*Figure 3C*; TP53<sup>wt</sup> FLCN<sup>KO</sup> refers to a TP53 positive FLCN knock out clone, see *Figure 3—figure supplement 1B*). Induction of GPNMB, RRAGD, SQSTM1, FNIP2 and other E-box targets was significantly ameliorated in all three FLCN<sup>NEG</sup> RPTEC clones by simultaneously knocking down TFE3 and its close family member TFEB (*Figure 3D*; siTFE3 only is shown in *Figure 3—figure supplement 1C*), an effect further confirmed in a TP53<sup>wt</sup> FLCN<sup>KO</sup> cell line shown in *Figure 8—figure supplement 1C*. We conclude that FLCN loss activates a specific category of TFE target genes in renal epithelial cells, independent of TP53 or karyotype, many of which function in autophagy and lysosome regulation.

Since the nutrient-sensing capabilities of mTOR had been associated with both FLCN and TFE3 activity in previous studies (*Baba et al., 2006*; *Bastola et al., 2013*; *El-Houjeiri et al., 2019*; *Hasumi et al., 2009*; *Hong et al., 2010b*; *Hudon et al., 2010*; *Petit et al., 2013*; *Takagi et al., 2008*), we next assessed FLCN-dependent changes in canonical mTOR signaling. However, phosphorylation levels of two direct mTORC1 targets, 4E-BP1 (as judged by its electrophoretic mobility shift) and S6 kinase (S6K_T389) were not changed by FLCN loss in RPTECs, while also AKT/PKB_S437 phosphorylation remained intact (*Figure 4A*). The dynamic subcellular localization of

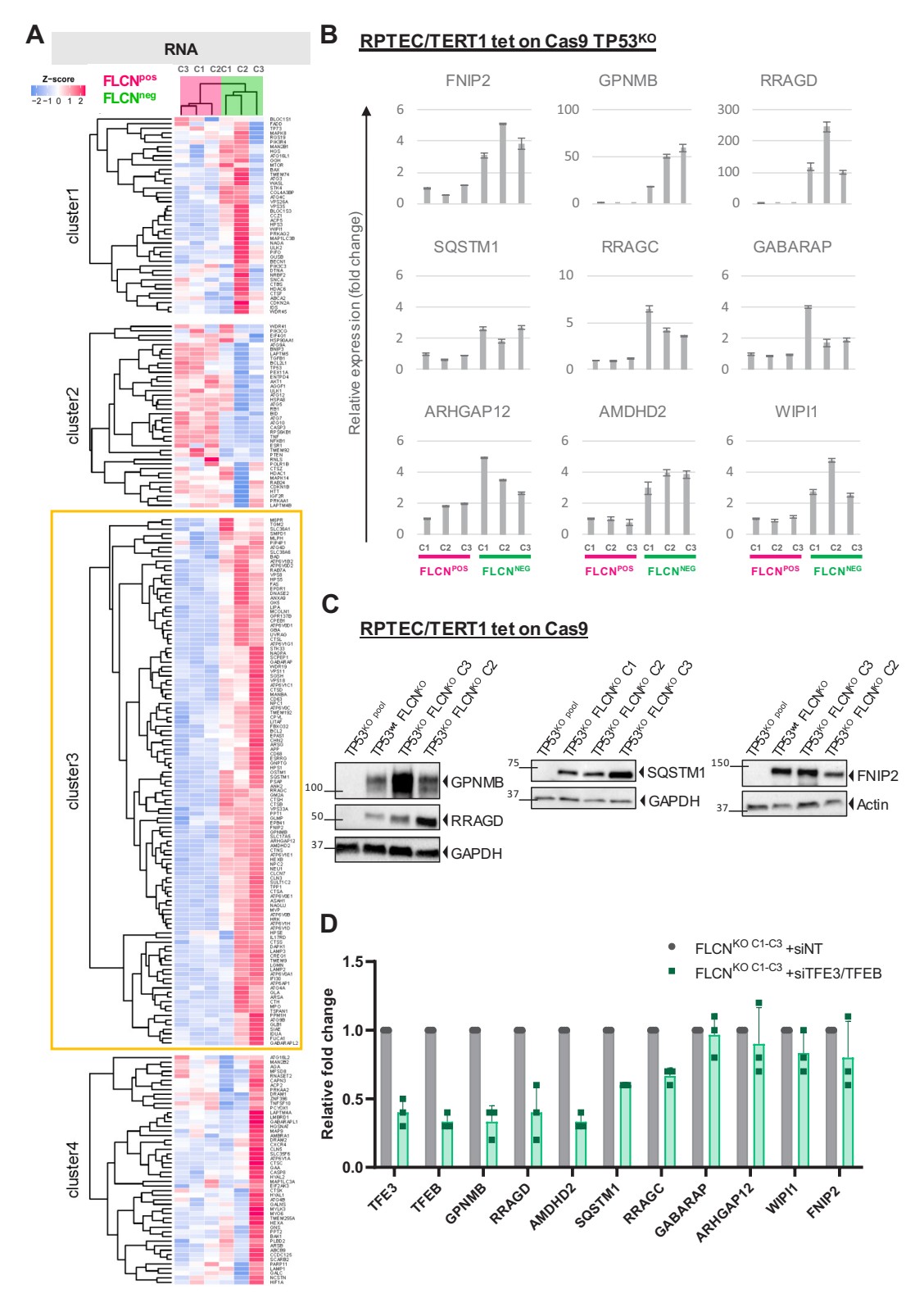

**Figure 3.** FLCN loss results in upregulation of subset of TFE target genes. (**A**) Heat map showing k-means Pearson correlation clustering of TMM-normalized RNAseq data of FLCN^pos versus FLCN^neg RPTECs. We analyzed published TFEB/TFE3 target genes. Yellow boxed cluster three shows the subset (n = 115) of TFEB/TFE3 targets upregulated in all three FLCN^NEG clones. (**B**) Upregulation of TFE target genes FNIP2, GPNMB, RRAGD, SQSTM1, RRAGC, GABARAP, ARHGAP12, AMDHD2, and WIPI1 in FLCN^NEG RPTECs. Results of three independent experiments with three technical

*Figure 3 continued on next page*

*Figure 3 continued*

replicates. To determine quantitative gene expression levels, data were normalized to the geometric mean of two housekeeping genes. See *Figure 3— source data 1* for raw qRT-PCR values and fold change calculations. (C) Western blots of RPTEC/TERT1 tet-on Cas9 cell lines. All FLCN[NEG] clones show strong induction of protein expression of TFE targets GPNMB, RRAGD, SQSTM1, and FNIP2. GAPDH and Actin were used as loading controls. Western blots were performed three times. (D) Knock down of TFE3/TFEB (10 nM siRNA, 72 hr) ameliorates the TFE expression gene signature induced by FLCN loss in three FLCN[NEG] clones. Expression levels were determined by qRT-PCR, normalized to siNT-treated clones and are representative of three independent experiments. To determine quantitative gene expression data levels were normalized to the geometric mean of two housekeeping genes. Also see *Figure 8—figure supplement 1C*. Effects of siTFE3 alone are shown in *Figure 3—figure supplement 1C*. See *Figure 3—source data 1* for raw qRT-PCR values and fold change calculations.

The online version of this article includes the following source data and figure supplement(s) for figure 3:

**Source data 1.** Raw qRT-PCR values and fold change calculations belonging to *Figure 3B and D* and *Figure —figure supplement 1C*.
**Figure supplement 1.** Comparative analyses of RPTEC FLCN[POS] vs. FLCN[NEG] cell line pairs and validations of TP53[WT] FLCN[KO] RPTEC cell line.

mTOR in response to starvation did not differ between FLCN[POS] and FLCN[NEG] as detected by immunofluorescent co-staining of mTOR and lysosomal marker LAMP2 (*Figure 4B*, FLCN[NEG] C3 is representative of the three FLCN[NEG] clones which all show normal mTOR dynamics). So, in contrast to several previous studies, but in line with a recent report (*El-Houjeiri et al., 2019*), we found no evidence for directly altered mTOR signaling or nutrient sensing in the absence of FLCN in renal tubular cells.

## An interferon (IFN) gene signature is induced in the absence of FLCN

To further identify the main biological processes influenced by FLCN expression, we performed Molecular Signatures Database (MSigDB) gene set enrichment analyses (GSEA) (*Subramanian et al., 2005*) on both RNA and protein data sets (pre-ranked list of p-values, classic ES). *Figure 5A* displays hallmark gene sets ranked by normalized enrichment score (NES) and significance (FDR is represented by the size of the dot). Gene sets significantly enriched in either RNA or protein (FDR < 0.05) are shown, with biological processes enriched in FLCN[NEG] indicated in green, and processes enriched in FLCN[POS] in pink. An overview of less significant (FDR > 0.05) hallmark gene sets is shown in *Figure 6—figure supplement 1A*. We found a higher representation of cell cycle related processes in FLCN[POS] cells, in both RNA and protein data (*Figure 5A*, *Figure 6—figure supplement 1A*, pink marks: E2F_TARGETS, G2M_CHECKPOINT, MITOTIC_SPINDLE, MYC_TARGETS). Growth curves confirmed that deletion of FLCN reduced proliferation of RPTEC cells significantly (p=8.31E-11; Table 1 and accompanying Figure in Materials and methods), an unexpected effect of tumor suppressor gene inactivation (*Figure 5B*). Other typical cellular processes and signal transduction cascades that mark FLCN[POS] RPTECs were MTORC1_SIGNALING, HYPOXIA, and TGF_BETA_SIGNALING. Interestingly, however, in FLCN[NEG] RPTECs, the immune-response-related hallmarks were highly significantly enriched in both RNA and protein data (*Figure 5A*, *Figure 6—figure supplement 1A*, IFN_GAMMA_RESPONSE, IFN_ALPHA_RESPONSE, COMPLEMENT). BinGO (*Maere et al., 2005*) gene ontology analyses of differential expression patterns in FLCN[NEG] RPTECs also revealed many overlapping immune and IFN response signature genes, including ISG15, IFIT1, IF16, MX1, OAS2, and STAT2 in both RNA and protein data (*Figure 5C,D*; orange circles show overlap). These findings indicate that the IFN response signature program is a key target of FLCN in renal epithelial cells.

## Two distinct transcriptional programs are strongly induced by FLCN loss

To specify how transcription is changed in FLCN[NEG] RPTEC, we used iRegulon (*Janky et al., 2014*), which prioritizes candidate regulatory transcription factors based on enriched promotor motifs upstream of the transcription start sites (TSS) (*Figure 6A*). The iRegulon analysis of distinct promotor motifs directing genes upregulated in FLCN[NEG] (n = 711, FDR < 0.05 and logFC > 2) are shown in *Figure 6B*. A similar overview is shown in *Figure 6C*, based on significantly upregulated proteins in FLCN[NEG] RPTEC (n = 498, p<0.05 and FC > 2).

Importantly, the majority of regulatory elements enriched in FLCN[NEG] RPTEC can be assigned to either the basic helix-loop-helix E-box motif group (e.g. regulated by TFE3/TFEB) or to the Interferon-Stimulated Response Element (ISRE) motif group (*Figure 6D*). In FLCN[POS] RPTECs, iRegulon

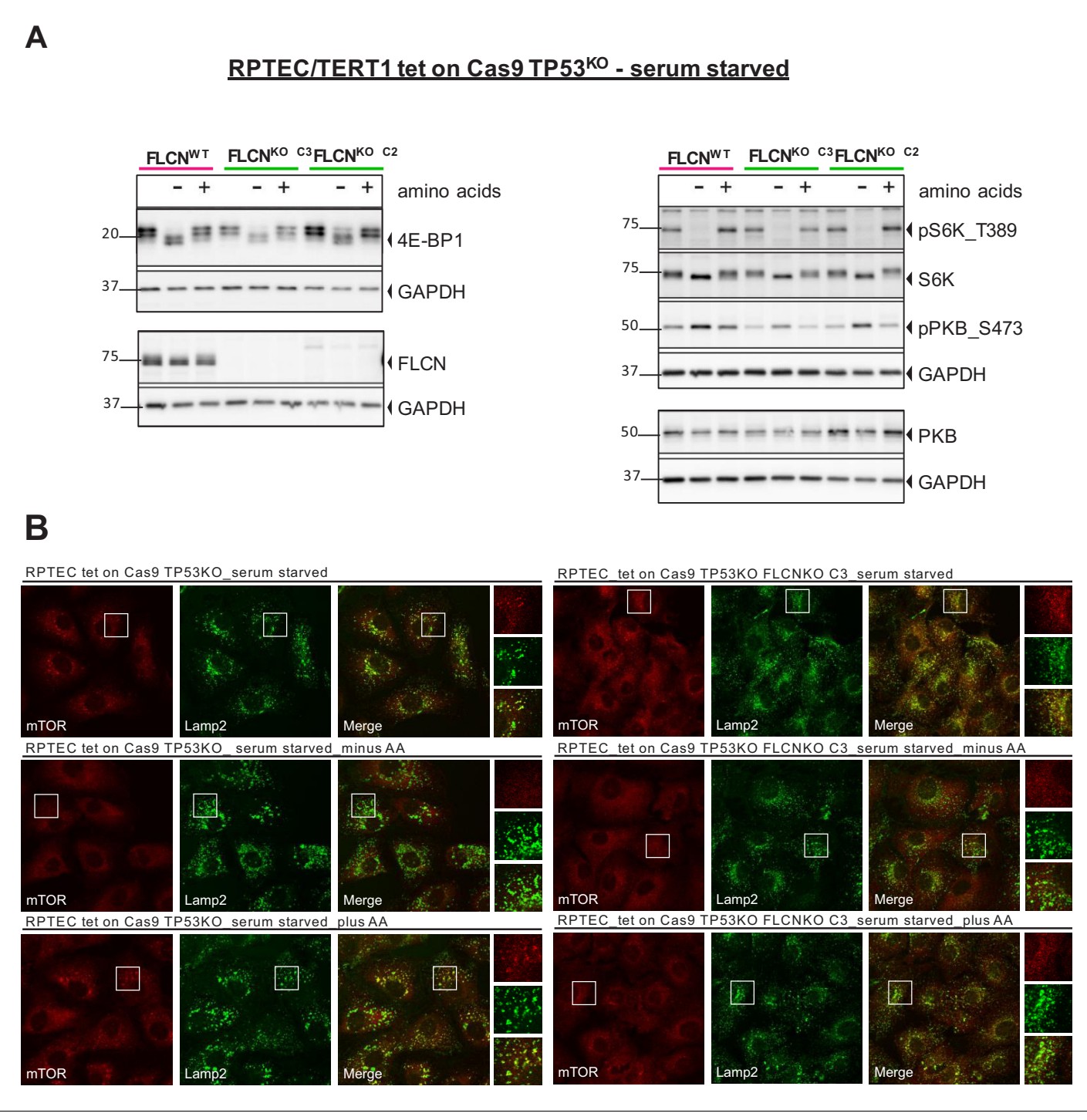

**Figure 4.** mTOR localization and signaling in response to starvation does not change upon FLCN loss in RPTEC. (**A**) To detect changes in canonical mTOR signaling, phosphorylation levels of S6 kinase (S6K_T389) and AKT/PKB (PKB_S473) and total protein levels of S6K, AKT/PKB, 4E-BP1 were assessed by western blot. Serum starved FLCN[POS] and FLCN[NEG] RPTEC cell lines with and without additional amino acids (AA) depletion were analyzed three times. (**B**) Immunofluorescence staining of mTOR and lysosomal marker LAMP2 show no FLCN dependent difference of mTOR localization in response to starvation. FCS = fetal calf serum, AA = amino acids. Staining of FLCN[NEG] RPTEC C3 is representative for three independent FLCN[NEG] clones.

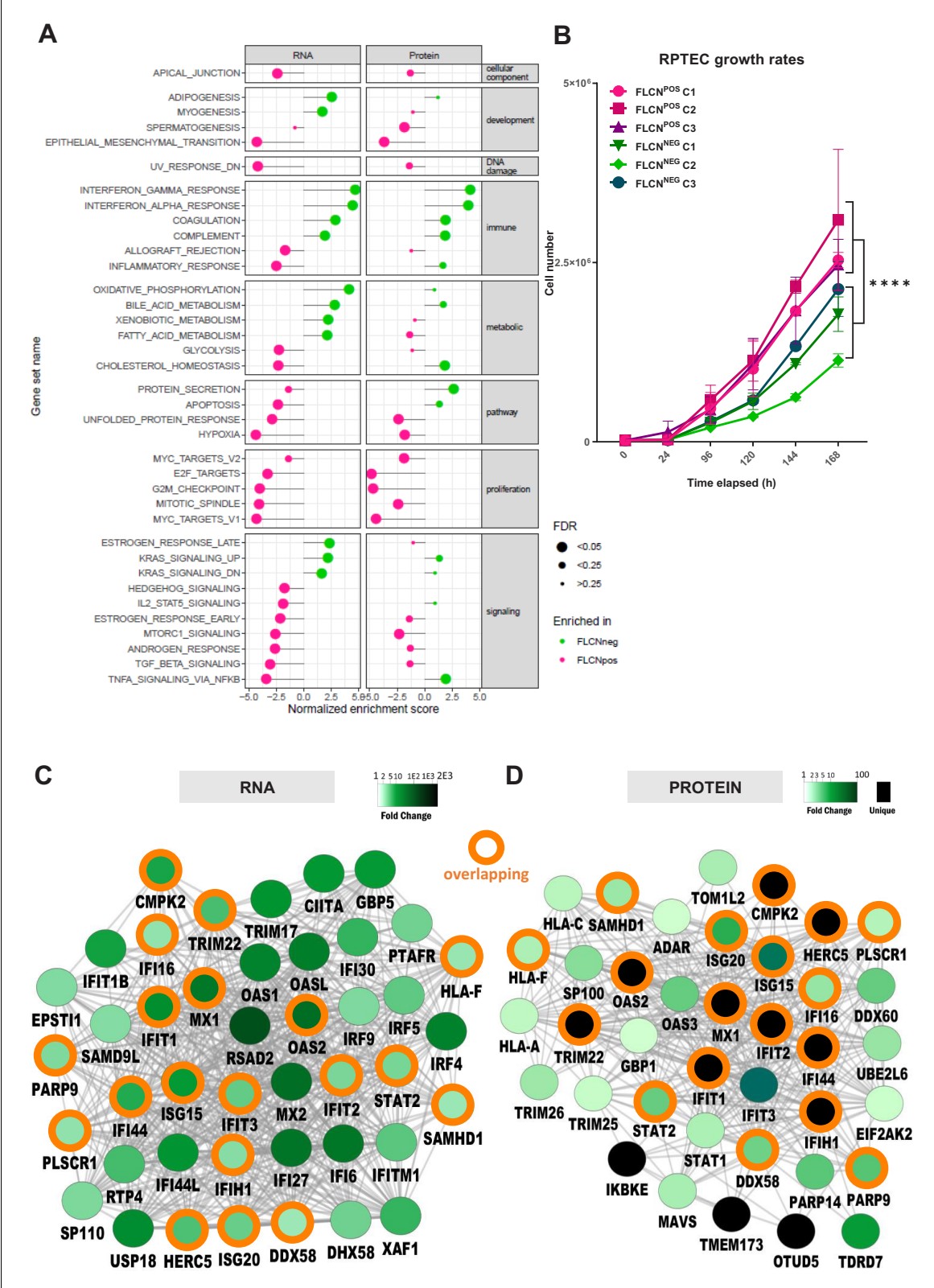

**Figure 5.** Gene set enrichment analysis reveals FLCN-dependent biological processes. (**A**) For Gene Set Enrichment Analysis (GSEA) genes or proteins were ranked based on p-values, with genes/proteins that are expressed significantly higher in FLCN[NEG] RPTECs shown on top of the list (hallmark gene sets, classic ES). Enriched hallmark gene sets are ranked by normalized enrichment score (NES). Gene sets enriched in FLCN[NEG] are shown in green and gene sets enriched in FLCN[POS] in pink. The size of the dot reflects the significance of the enrichment (FDR=false discovery rate). Only biological

*Figure 5 continued on next page*

Figure 5 continued

processes that were significant in either RNA and/or protein data are depicted in this Figure. An extended version with all identified gene sets is shown in **Figure 6—figure supplement 1A**. (B) FLCN^NEG RPTECs grow significantly slower (p=8.31E-11) when compared to FLCN^POS RPTEC. Cell lines were seeded in equal densities and total cell number was counted for 7 consecutive days. Results shown are representative for two independent experiments. (C) Gene Ontology (biological processes, BinGO) analysis of mRNAs higher expressed in FLCN^NEG RPTECs reveals highly overlapping (orange circles) clusters of immune- and interferon-response-related genes between both data sets. Shade of green nodes represents fold change. (D) Gene Ontology (biological processes, BinGO) analysis of proteins higher expressed in FLCN^NEG RPTECs reveals highly overlapping (orange circles) clusters of immune and interferon response related genes between both data sets. Shades of green nodes represent different levels of fold change. Black nodes indicate uniquely detected proteins in FLCN^NEG RPTEC.

analyses did not identify regulatory elements shared in both RNA and protein data (**Figure 6—figure supplement 1B**). Upregulation of E-box or ISRE motif-dependent genes induced by FLCN loss in our RNAseq and proteomics analyses are shown in **Figure 6E–H**. Significant induction of the ISRE targets MX1, ISG15, IRF9, IFIT1, STAT1, and STAT2 next to TFE3 and E-box targets GPNMB, RRAGD, FNIP2, CTSD, and SQSTM1 were validated by qPCR in **Figure 6—figure supplement 1C**. We then analyzed protein expression levels compared to normal human kidney lysates using mass spectrometry of two independent BHD kidney tumors (**Figure 6—figure supplement 1D and E**). Although full interpretation of these results awaits the analysis of a larger tumor sample size, we observed elevated expression of both the ISRE as E-box associated genes in FLCN^NEG BHD tumors (**Figure 6I**). Based on these results, we conclude that FLCN loss upregulates two major gene classes: TFE3/TFEB regulated E-box targets and IFN-associated ISRE targets. Our integrated analysis reveals these genes and their protein products as candidate positive biomarkers for FLCN loss in BHD-related kidney cancer.

## FLCN-FNIP1/2 loss upregulates STAT2 in a cell-type-specific manner

FLCN acts in a protein complex with FNIP1 and FNIP2 (**Baba et al., 2006**; **Hasumi et al., 2015**; **Hasumi et al., 2008**). When FNIP1 and FNIP2 are inactivated simultaneously, mice develop kidney cancer (**Hasumi et al., 2015**). To investigate whether deletion of both FNIP1 and FNIP2 in RPTEC had a molecular effect similar to that of FLCN loss, we created a FNIP1/FNIP2^NEG RPTEC cell line (**Figure 7—figure supplement 1A**) and analyzed gene induction. This confirmed that upregulation of ISRE or E-box motif genes is specifically connected to inactivation of the FLCN-FNIP1/FNIP2 axis (**Figure 7A**). Furthermore, TFE3 localized to the nucleus upon FNIP1/2 loss (**Figure 7—figure supplement 1B**).

Having validated a new gene program targeted by the FLCN-FNIP1/2 complex in RPTECs, we next determined whether FLCN-dependent control of ISRE target genes occurred in cells of different tissue origin, too. Using a similar approach as in RPTEC, we created a FLCN^NEG retinal pigment epithelial cell line (RPE1/TERT tet-on Cas9 TP53^KO **Benedict et al., 2020**; **Figure 7—figure supplement 1C**). Quantitative RT-PCR analyses revealed that, strikingly, ISRE and E-box associated genes were not induced in RPE1 cells by FLCN loss (**Figure 7B**). RRAGD was the only exception yet was induced ~10 times less strongly in RPE1 as compared to RPTEC cells. These results indicate that the two main gene induction programs directed by FLCN-FNIP1/2 are renal specific.

To identify the dominant signature of regulatory elements responding to FLCN loss in RPTEC, we further looked into the shared hits of RNA and protein data sets. Upon selecting the most significantly overlapping effects (FDR < 0.01, n = 181) (**Figure 7C**), iRegulon analysis of differential RNAs overlapping with differential proteins (r > 0.8, n = 49) upon FLCN loss predicted STAT1 and STAT2-binding motifs as the most enriched upstream regulatory elements (**Figure 7D**).

To prove that these upregulated gene programs were truly FLCN dependent, we complemented FLCN^NEG RPTEC C2 by re-introducing FLCN (**Figure 7—figure supplement 1D**). FLCN re-expression in FLCN^NEG RPTEC C2 restored regulation of TFE3 localization and nutrient sensing (**Figure 7—figure supplement 1E**). Moreover, re-introducing FLCN completely reverted the ISRE expression phenotype (**Figure 7E**). To confirm the roles of STAT1/2, we knocked them down using siRNAs in FLCN^NEG RPTEC C2. Quantitative RT-PCR showed that upregulation of the ISRE-associated gene program was entirely STAT1/2-dependent (**Figure 7E**). In addition, immunoblotting subcellular fractions showed higher protein levels of both STAT1 and STAT2 in FLCN^NEG cells (**Figure 7F**).

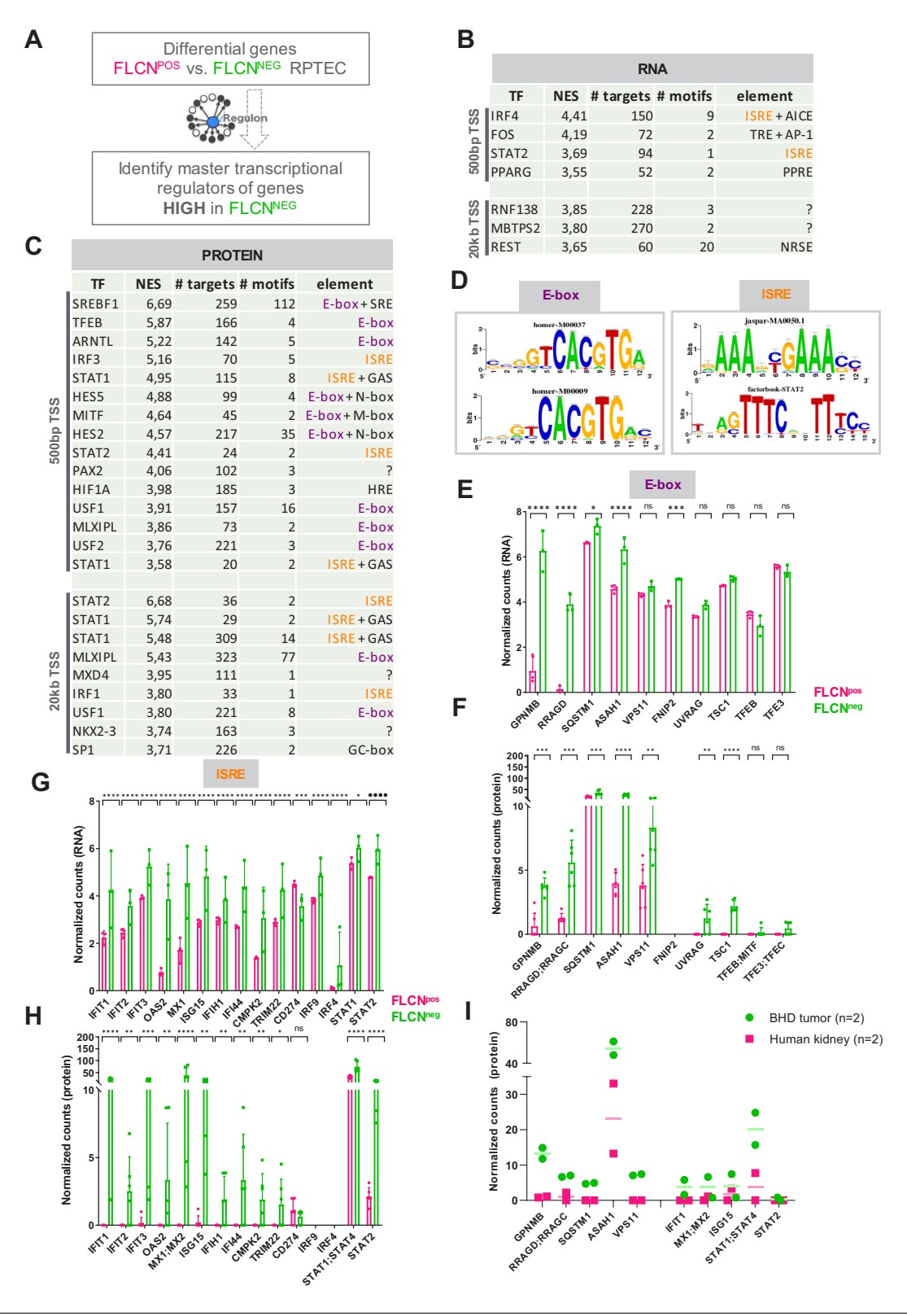

**Figure 6.** Identification of regulatory elements activated by FLCN loss in RPTEC and BHD tumors. (A) Identification of transcriptional regulatory elements associated with loss of FLCN expression. Regulons were identified by iRegulon (*Janky et al., 2014*), using an input a list of differential expressed genes (*Figure 3*). (B) Upstream regulons enriched in FLCN[NEG] RPTEC based on significantly upregulated genes derived from our transcriptomic data set (n = 711, FDR < 0.05 and logFC > 2). Transcription factors with normalized enrichment scores (NES) higher than 3.5 are shown,
*Figure 6 continued on next page*

Figure 6 continued

together with detected number of targets, motifs, and elements. ISREs are highlighted in orange. Upper part shows motifs enriched 500 bp upstream from transcription start site (TSS), lower part shows motifs enriched 20 kb around TSS. (C) Upstream regulons enriched in FLCN^NEG RPTEC based on significantly upregulated proteins derived from our proteomic data set (n = 498, p<0.05 and FC > 2). Transcription factors with normalized enrichment score (NES) higher than 3.5 are shown, together with number of targets, motifs, and elements detected. ISREs are highlighted in orange and E-boxes in purple. STAT1 appears twice due to the fact that iRegulon ranks this transcription factor to be the most likely upstream regulator for two sets of targets genes, containing slightly different ISRE-motifs 20 kb upstream from the TSS. (D) Two major enriched motif elements detected in iRegulon analysis of genes upregulated in FLCN^NEG RPTEC. Regulons can be assigned to E-box (in purple) or ISRE (in orange) motif group. (E) Bar graphs of RNA expression levels of genes associated with an E-box motif, derived from RPTEC transcriptomic data set. FLCN^POS values are shown in pink and FLCN^NEG values are shown in green. Significant p-values are indicated as *≤0.05, **≤0.01, ***≤0.001, ****≤0.0001. (F) Bar graphs of protein expression levels of genes associated with an E-box motif, derived from RPTEC proteomic data set. FLCN^POS values are shown in pink and FLCN^NEG values are shown in green. FNIP2 peptides were not detected in our proteomic experiment and therefore absent in the bar graph. Significant p-values are indicated as *≤0.05, **≤0.01, ***≤0.001, ****≤0.0001. (G) Bar graphs of RNA expression levels of genes associated with an ISRE motif derived from RPTEC transcriptomic data set. FLCN^POS values are shown in pink and FLCN^NEG values are shown in green. Significant p-values are indicated as *≤0.05, **≤0.01, ***≤0.001, ****≤0.0001. (H) Bar graphs of protein expression levels of genes associated with an ISRE motif derived from RPTEC proteomic data set. FLCN^POS values are shown in pink and FLCN^NEG values are shown in green. IRF9 and IRF4 peptides were not detected in our proteomic experiment and therefore absent in the bar graph. Significant p-values are indicated as *≤0.05, **≤0.01, ***≤0.001, ****≤0.0001. (I) Dot plot of protein expression levels of genes associated with an E-box (left) or ISRE motif (right) derived from BHD kidney tumor proteomic data sets (see *Figure 6—figure supplement 1D and E*), as compared to normal kidney tissue. FLCN^POS values are shown in pink and FLCN^NEG values are shown in green. STAT2 levels were below detection levels in these protein extracts.

The online version of this article includes the following source data and figure supplement(s) for figure 6:

**Figure supplement 1.** Extended GSEA and iRegulon analysis of FLCN loss in RPTEC and BHD tumors.

**Figure supplement 1—source data 1.** Raw qRT-PCR values and fold change calculations belonging to *Figure 6—figure supplement 1C*.

Importantly, nuclear STAT2 was exclusively bound to chromatin in the absence of FLCN (*Figure 7G*) identifying STAT2 as a key target of FLCN.

Canonical IFN signaling follows IFNα or IFNγ stimulation of IFN receptors, resulting in auto-phosphorylation of Janus Kinases 1/2 (JAK1/JAK2) and Tyrosine Kinase 2 (TYK2). As a consequence, STAT1 and STAT2 are phosphorylated enabling formation of (homo)dimers or the ISGF3 complex (composed of STAT1, STAT2, and IRF9) which translocate to the nucleus to initiate transcription of ISRE genes (*Majoros et al., 2017*). To understand how STAT2 is activated upon FLCN loss, we measured IFN levels in supernatant of FLCN^NEG RPTEC cell lines, using a flow-cytometry-based cytometric bead array (CBA) or an enzyme-linked immunosorbent assay (ELISA). However, we did not detect any secreted IFNγ or IFNα, indicating that autocrine stimulation of IFN signaling did not cause the upregulated ISRE gene program or chromatin binding of STAT2 in FLCN^NEG RPTEC (*Figure 7—figure supplement 1F*). Also, we did not detect IFNA, IFNB and IFNG expression, nor differential expression of the IFN receptors (IFNAR1, IFNAR2, IFNGR1 and IFNGR2) in FLCN^NEG RPTEC (*Figure 7—figure supplement 1G*). Finally, in a recent phospho-proteomic analysis of our FLCN^NEG cell lines (not described in this paper), loss of FLCN did not lead to phosphorylation of upstream kinases JAK2 and TYK, or STAT2 itself. Lack of induced STAT1 phosphorylation as detected by immunoblot further confirmed that FLCN^NEG cells do not activate the IFN receptor (*Figure 7—figure supplement 1H*). Together, these results show that FLCN loss in RPTEC leads to a non-canonical, IFN-independent activation of unphosphorylated STAT1 and STAT2.

## FLCN loss counteracts TFE3-induced hyperproliferation

As the experiments described above confirm our hypothesis that both TFE3 and STAT2 are FLCN targets, we started to investigate their potential contribution to renal cell transformation in vitro. We aimed to compare the effect of TFE3 activation resulting from FLCN loss to that of an active TFE3 gene fusion that is constitutively nuclear. Xp11 translocation RCC is a rare subtype of kidney cancer associated with various TFE3, *TFEB*, or MITF gene fusions, resulting in oncogenic, nuclear forms of these transcription factors (*Caliò et al., 2019*). This type of RCC behaves remarkably aggressively, with poor progression-free survival rates (*Lee et al., 2018*). One common fusion partner for TFE3 is the DNA-binding splicing factor SFPQ (*PSF*), which we expressed in RPTEC. For comparison, we created a new, diploid FLCN knock-out cell line by a recently improved gRNA and Cas9 delivery protocol (*Figure 8—figure supplement 1A and B*). In both SFPQ-TFE3 and FLCN^KO RPTEC cell lines, we confirmed high expression of TFE targets GPNMB, RRAGD, FNIP2, and WIPI1 (*Figure 8A*). TFE3 or

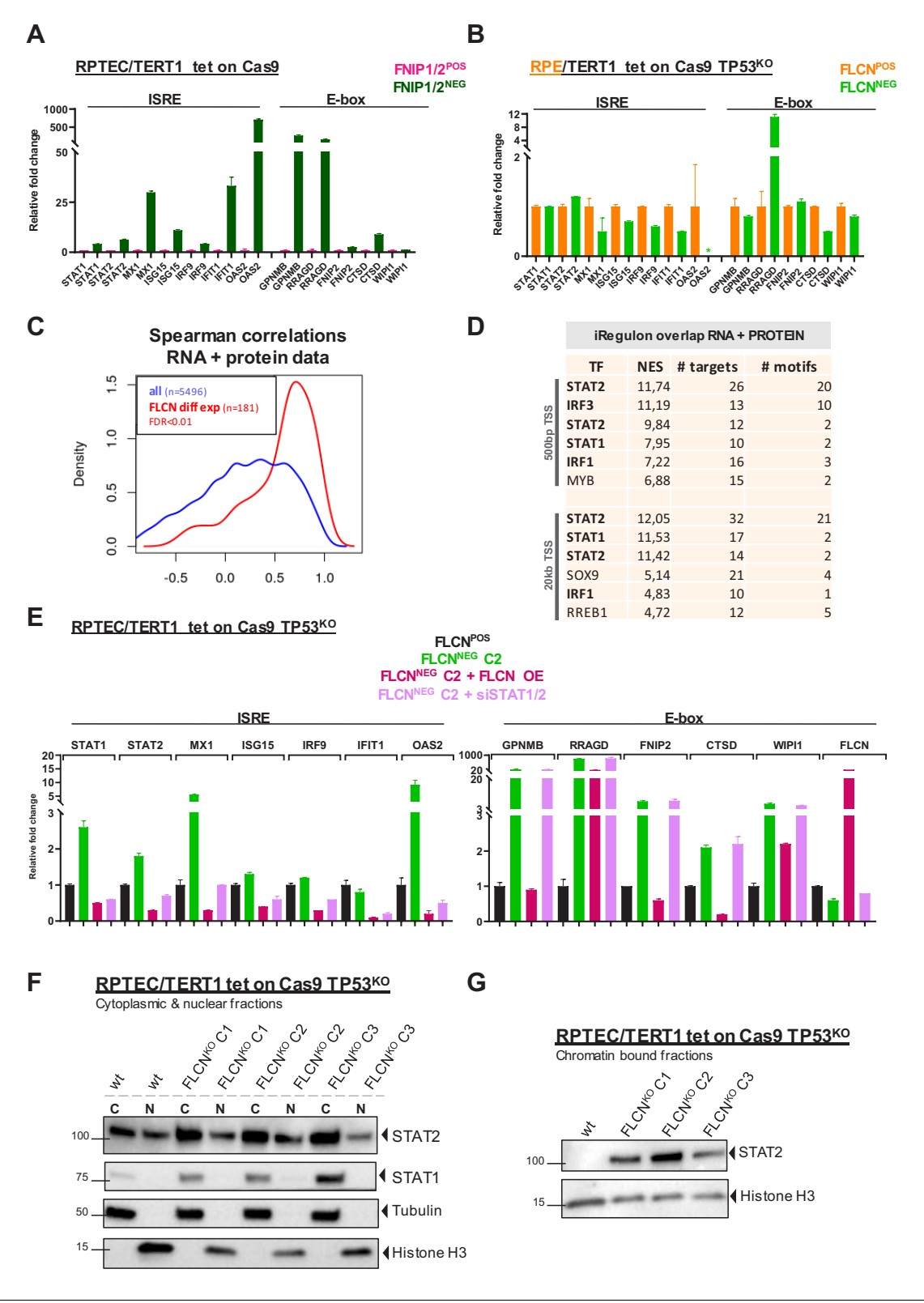

**Figure 7.** Inactivation of the FLCN-FNIP1/2 axis activates STAT2 in renal cells. (**A**) qRT-PCR levels of genes with ISRE or E-box motif in FNIP1[POS]/FNIP2[POS] and FNIP1[NEG]/FNIP2[NEG] RPTEC cells reveal that the identified FLCN-dependent gene signature is also induced upon loss of FLCN interacting proteins FNIP1 and FNIP2. Results shown are representative for two independent experiments with three technical replicates. To determine quantitative gene expression data levels were normalized to the geometric mean of two housekeeping genes. See *Figure 7—source data 1* for raw

*Figure 7 continued on next page*

*Figure 7 continued*

qRT-PCR values and fold change calculations. (B) qRT-PCR levels of genes with ISRE or E-box motif in FLCN^POS and FLCN^NEG retinal pigment epithelial cells (RPE/TERT1 tet on Cas9 TP53^KO) reveal that the identified FLCN dependent gene signature is absent in an epithelial cell type of another tissue origin. Results shown are representative for two independent experiments. To determine quantitative gene expression data levels were normalized to the geometric mean of two housekeeping genes. *OAS2 level in FLCN^NEG RPE was too low to detect using qRT-PCR. See *Figure 7—source data 1* for raw qRT-PCR values and fold change calculations. (C) Spearman correlation analysis reveals overlapping FLCN-dependent RNA and protein data. FLCN differential mRNAs and proteins (FDR < 0.01, n = 181, red line) showed a higher correlation than the overlap of all identified mRNAs and proteins in our datasets (blue line). Statistical methods are described in Materials and methods section. (D) iRegulon analysis of differentially expressed genes (FDR < 0.01) with highest correlation with differentially expressed proteins (r > 0.8, n = 49) reveal STAT1, STAT2, IRF1, and IRF3 as most obvious upstream transcriptional regulators. Only regulons displaying normalized enrichment scores (NES) > 4.5 are shown. STAT2 appears twice due to the fact that iRegulon ranks this transcription factor to be the most likely upstream regulator for two sets of targets genes, containing slightly different ISRE-motifs upstream from the transcription start site (TSS). (E) Reintroducing FLCN (overexpression, OE) or siRNA-mediated knock down of STAT1/STAT2 (10 nM, 72 hr) revert the IFN expression gene signature induced by FLCN loss in RPTEC FLCN^NEG C2. FLCN OE also lowers the enhanced expression of E-box-associated target genes but knock down of STAT1/2 has no effect on E-box-associated genes. Expression levels were determined by qRT-PCR and are representative of two independent experiments. To determine quantitative gene expression data levels were normalized to the geometric mean of two housekeeping genes. See *Figure 7—source data 1* for raw qRT-PCR values and fold change calculations. (F) Western blots of subcellular fractionated samples show higher expression of STAT1 and STAT2 in FLCN^NEG RPTEC as compared to FLCN^POS RPTEC. STAT2 was also detected in both cytoplasmic and nuclear fractions. Tubulin and histone H3 levels were used as loading control and to distinguish each fraction (N=nuclear, C=cytoplasmic). Results shown are representative of two independent fractionations. (G) Western blot of subcellular fractionated samples shows enhanced STAT2 DNA binding in FLCN^NEG RPTEC. Results shown are representative of three independent fractionations.

The online version of this article includes the following source data and figure supplement(s) for figure 7:

**Source data 1.** Raw qRT-PCR values and fold change calculations belonging to 7A, 7B and 7E.
**Figure supplement 1.** Validation of FLCNs role in the IFN response in additional cell models.

the constitutively active SFPQ-TFE3 fusion protein are both bound to chromatin fractions of FLCN^KO RPTEC or SFPQ-TFE3 RPTECs, respectively (*Figure 8B*). Knock down of TFE3/TFEB in the new TP53 wild-type FLCN^KO RPTEC cell line confirmed that TFE is required for the E-box expression program induced by FLCN loss (*Figure 3D*, *Figure 8—figure supplement 1C*).

Importantly, SFPQ-TFE3 expression did not induce IFN response genes in RPTECs, showing that the IFN gene induction signature is a specific effect of FLCN loss (*Figure 8A*). Concordantly, STAT2 recruitment to chromatin was enhanced in the absence of FLCN but not detected after expression of SFPQ-TFE3 (*Figure 8B*).

In agreement with the known growth inhibitory effects of the IFN stimulated gene program, but unexpected considering that FLCN is a tumor suppressor gene, FLCN loss reduced RPTEC colony formation and slowed cellular proliferation, regardless of TP53 status (*Figure 8C* upper panels, *Figure 5B*). Indeed, re-introducing FLCN expression in FLCN^KO RPTEC rescued cellular proliferation (*Figure 8D*). Reversely, the absence of FLCN dominantly repressed the hyper-proliferative effects of an active, oncogenic TFE3 fusion protein in RPTEC, similar to the growth reduction observed after treating SFPQ-TFE3 RPTECs with 100 IU/ml IFNγ (*Figure 8C* lower panels, *Figure 8E*). The growth inhibitory effect of knocking out FLCN in SFPQ-TFE3 RPTECs also correlated with strong induction of ISRE genes (*Figure 8—figure supplement 1D*). In conclusion, these results show that FLCN loss induces a STAT2-mediated IFN signature that results in growth inhibition, which counteracts the hyperproliferative effects of constitutive activation of TFE3 (*Figure 8—figure supplement 1E*). This indicates that, next to its growth stimulatory effects on cell proliferation via TFE3 activation, loss of FLCN also has a growth suppressive effect executed by STAT2 activation in renal tubular cells. STAT2 activation may also contribute to a pro-oncogenic state by contributing to an inflammatory response (*Figure 8—figure supplement 1E*). Determining the role of STAT2 activity in tumorigenesis will require further analysis of bio-markers and immune-infiltrates in a large set of BHD tumors.

## Discussion

FLCN, together with its binding partners FNIP1 and FNIP2, forms a regulatory protein complex found in species ranging from yeast to humans (*Nookala et al., 2012*; *Pacitto et al., 2015*; *Zhang et al., 2012*). Pleiotropic effects resulting from its mutation in different model systems have hampered a clear understanding of the biological function of the FLCN-FNIP1/2 axis. At the organismal level, FLCN also plays different roles in different tissues. This is already apparent from the clinical

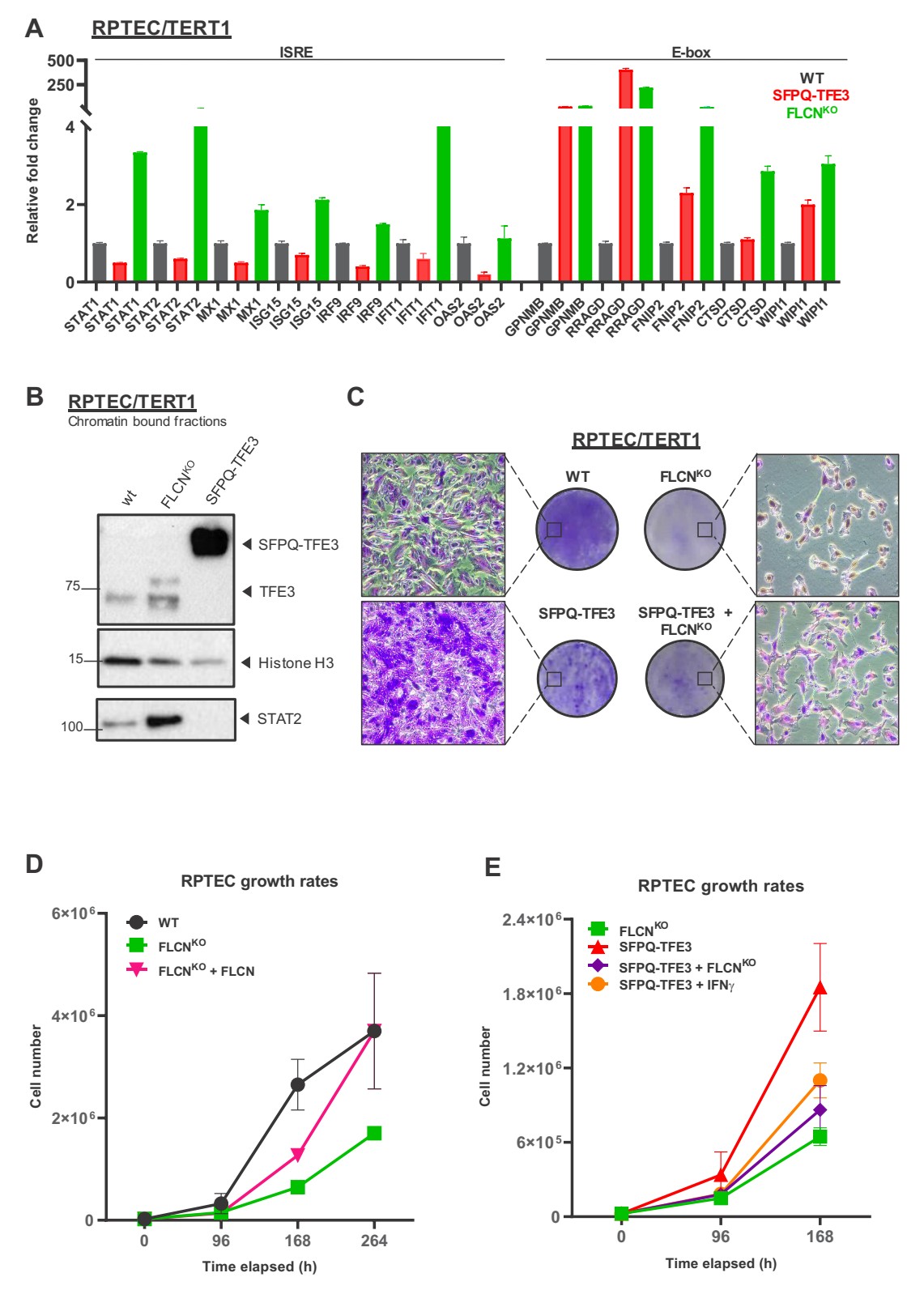

**Figure 8.** FLCN loss induces an interferon signature which counteracts growth promoting effects of active TFE3 in renal tubular cells. (**A**) Expression of a constitutively active SFPQ-TFE3 fusion protein in RPTEC results in upregulation of E-box-associated targets but does not induce enhanced expression of ISRE-associated genes. FLCN$^{KO}$ RPTEC cells show both upregulation of E-box and ISRE-associated genes. Expression levels were determined by qPCR and are representative of two independent experiments. To determine quantitative gene expression data, levels were normalized to the

*Figure 8 continued on next page*

*Figure 8 continued*

geometric mean of two housekeeping genes. See *Figure 8—source data 1* for raw qRT-PCR values and fold change calculations. (**B**) Western blots of subcellular fractions show enhanced binding of TFE3 to DNA in FLCN[NEG] RPTEC and SFPQ-TFE3 RPTEC. STAT2 DNA-binding was enhanced upon FLCN loss but reduced by SFPQ-TFE3 over-expression in RPTEC. STAT2 was blotted on separate blots of the same lysates. Histone H3 levels were used as loading control and as marker for chromatin fraction. Western blot was performed two times, using independent fractionations. (**C**) Colony formation assays show that loss of FLCN in wild-type or SFPQ-TFE3 RPTEC reduces colony outgrowth. SFPQ-TFE3 RPTEC show more colonies than wild type RPTEC after 10 days. Insets show bright field images (×20 magnification). Cells were seeded in three technical replicates and experiment was performed twice. (**D**) Loss of FLCN in RPTEC results in slower growth, which is reverted when FLCN expression is restored by over-expression. Cell lines were seeded in equal densities and total cell numbers were counted three times within 11 days. Results shown are representative for two independent experiments. (**E**) Treatment with IFNγ (100IU/ml) or combining FLCN[KO] in SFPQ-TFE3 RPTEC results in growth inhibition. Cell lines were seeded in equal densities and total cell number was counted twice within 7 days. Results shown are representative for two independent experiments. The growth curve of FLCN[KO] RPTEC (**D**) is added for comparison.

The online version of this article includes the following source data and figure supplement(s) for figure 8:

**Source data 1.** Raw qRT-PCR values and fold change calculations belonging to *Figure 8A* and *Figure 8—figure supplement 1C and D*.
**Figure supplement 1.** Creation and validation of FLCN[NEG] and SFPQ-TFE3 RPTECs.

manifestations in BHD syndrome, varying from fibrofolliculomas of the skin, to pulmonary cysts with increased risk for pneumothorax, and renal cysts associated with increased cancer risk (*Birt et al., 1977*; *Houweling et al., 2011*; *Nickerson et al., 2002*; *Schmidt et al., 2005*; *Zbar et al., 2002*).

While manifestations in the skin and lung in BHD probably reflect an effect of FLCN haplo-insufficiency, kidney tumorigenesis in BHD carriers starts by complete functional inactivation of the remaining wild type FLCN allele (*van Steensel et al., 2007*; *Vocke et al., 2005*). Here, we modeled the molecular and cellular effects of FLCN inactivation in the cell type most relevant for kidney tumorigenesis. The RPTEC/TERT1 cell line is widely accepted as an appropriate in vitro model system for human kidney function (*Aschauer et al., 2015*). Nevertheless, it is important to emphasize that many of our conclusions are based on independent clones derived from a single human cell line. Here, we observed that loss of FLCN had three main effects in RPTEC/TERT1: (1) severely reduced cellular proliferation, regardless of TP53 activity or alterations in karyotype; (2) nuclear accumulation and activation of TFE3, concomitant with TFE3/TFEB-dependent upregulation of a specific set of E-box genes linked to autophagy and lysosomal control; (3) IFN-independent upregulation and chromatin binding of STAT2, activating a gene program of typical IFN response genes, possibly in cooperation with STAT1. Upregulated genes besides STAT1 and STAT2 include MX1, IFIT1, ISG15, IRF9, and OAS2. We found the combination of these two master gene programs to be a renal-specific response to FLCN inactivation, also observed after knocking out FNIP1 and FNIP2. Re-expressing FLCN reverted these effects, proving that they are specific to FLCN loss.

How these gene expression programs contribute to oncogenesis needs to be resolved, but TFE3 activation was clearly sufficient to promote uncontrolled, enhanced proliferation and loss of contact inhibition in RPTEC cells (e.g. *Figure 8*). Knocking out *FLCN* in the context of active TFE3 slowed cellular proliferation to the level observed after IFNγ treatment. The upregulated IFN signature could thus explain why loss of the FLCN tumor suppressor, paradoxically, represses cellular proliferation. In relation to cancer, slow growth induced by FLCN loss and STAT2 upregulation may form a barrier to TFE3-driven renal tumorigenesis in BHD patients. On the other hand, it is possible that the activation of the IFN program is associated with a pro-oncogenic inflammatory response. The TFE3 and IFN signature programs are connected in this respect: TFE3/TFEB upregulate cytokines in macrophages that elicit an innate immune response linked to pathogen resistance (*El-Houjeiri et al., 2019*). In our RPTEC FLCN[NEG] cell models however, we observed a clear downregulation in secreted cytokine IL-8 and no consistent changes in IL-6 expression (data not shown), suggesting that, at least in renal cells, the IFN program induced by FLCN inactivation is not connected to inflammation per se.

Preliminary analysis of BHD tumor material showed the presence of immune cells at tumor margins (unpublished findings) and upregulation of ISRE genes (*Figure 6I*). Collectively our data lead to the hypothesis that renal tumorigenesis in BHD patients could follow two different paths: either pro-inflammatory effects of specific E-box and/or ISRE genes are further aggravated by secondary mutations during tumor evolution, or growth-suppression by IFN signature genes, which is dominant over the effects of TFE3 activation, is gradually lost by additionally acquired mutations thus leading to

TFE3-driven tumor progression. This suggests that certain IFN response genes which strongly reduce proliferation form a deeper tumor suppressive layer protecting against uncontrolled proliferation in the absence of sufficient FLCN expression. This hypothesis fits with the observation that renal tumors specifically driven by TFE3 activation behave more aggressively as compared to slowly growing BHD tumors. Speculatively, BHD tumors fall into two mutually exclusive classes, with either high or low IFN signatures and/or STAT2 expression depending on their growth properties.

From our results, it is clear that FLCN loss promotes STAT2 binding to chromatin but we did not resolve a clear mechanism by which FLCN loss promotes these changes. Because TFE3 activation is not sufficient to induce STAT2 or upregulate ISRE genes, and reciprocally STAT1/2 siRNA does not downregulate the E-box genes, we conclude that the two gene activation programs are separate effects of FLCN loss. We considered that FLCN might affect STAT2 protein stability, in a similar manner as observed for a number of viral proteins that bind directly to STAT2 (*Grant et al., 2016*; *Morrison et al., 2019*) but found no evidence for direct complex formation between STAT2 and FLCN. Also, the protein half-life of STAT2 was similar in the absence of FLCN and no shift was observed in STAT2 nucleo-cytoplasmic distribution (data not shown).

With respect to candidate TFE3 target genes that drive renal epithelial cell transformation, it is clear that in the absence of FLCN, particularly GPNMB, RRAGD, ASAH1 and FNIP2, the latter in an apparent feedback mechanism, are strongly upregulated. In renal cells lacking FNIP1 and FNIP2, GPNMB and RRAGD are also very strongly induced. This illustrates the potential value of GPNMB and RRAGD as positive biomarkers for FLCN inactivation such as in BHD tumors.

GPNMB is a transmembrane protein frequently upregulated in a wide variety of tumors, including lung and renal cancer, yet it is unclear whether GPNMB overexpression in itself is tumorigenic (*Taya and Hammes, 2018*). Importantly, GPNMB can be targeted therapeutically using antibody-drug conjugates which are in clinical trials for cancer therapy, such as *glembatumumab vedotin* (*Rose et al., 2017*), providing an entry point for the evaluation of *glembatumumab* in the treatment of BHD tumors. Furthermore, RRAGD is a candidate oncogenic target gene of TFE3, previously associated with loss of FLCN (*Di Malta et al., 2017*; *Tsun et al., 2013*). Recently, *Napolitano et al., 2020* described TFEB to be the main driver of kidney abnormalities in a BHD mouse model. Their results show that TFEB is phosphorylated by mTORC1 in substrate-specific mechanism that is mediated by Rag GTPases. FLCN is a key regulator of Rag GTPases but despite the fact that we found clear upregulation of both RagC and RagD, we found no evidence for mTORC1 hyperactivation in our human in vitro model system. There might be crucial differences in renal tumorigenesis between mice and humans, where mTORC1 activation may occur at a later stage in oncogenic transformation of FLCN[NEG] renal epithelial cells.

In addition to the upregulated TFE3 and ISRE programs, we observed that several genes were downregulated by FLCN loss. GSEA analyses of downregulated genes showed overlapping biological processes but iRegulon analyses failed to reveal a clear common upstream transcriptional regulator of these genes in mRNA and protein data. Taken together however, the repository of FLCN target genes and proteins presented here provides a clear basis for further investigations into specific roles in kidney cancer and other BHD-related symptoms. This could facilitate the discovery of biomarkers for early-stage tumorigenesis and therapeutic strategies to prevent or treat RCC metastases in BHD patients.

## Materials and methods

### Cell culture

Renal proximal tubular epithelial cells (RPTEC/TERT1, ATCC CRL-4031) were maintained in DMEM/F12 (Gibco, Life Technologies, Thermo Fisher Scientific Inc, Waltham, Massachusetts, US) according to the manufacturer's protocol with addition of 2% fetal bovine serum (FBS, Gibco). To maintain the selective pressure for immortalization 0.1 mg/ml G418 Sulfate (Calbiochem, Merck, Darmstadt, Germany) was added. Cell lines were cultured in a humidified atmosphere at 37°C and 5% $CO_2$. Retinal pigment epithelial cells (RPE-1/hTERT, ATCC CRL-4000) were maintained in DMEM (Gibco, Life Technologies) with addition of 8% FBS and 1 mM Sodium Pyruvate (Gibco, Life Technologies). The generation of used RPE1-hTERT tet-on Cas9 TP53[KO] cells was described earlier (*Benedict et al., 2020*). Both cell lines were obtained recently from ATCC and experiments were performed

exclusively with low-passage cell lines which were regularly tested to exclude Mycoplasma infections; follow-up authentication was performed on the basis of functional assays, gene expression patterns and cellular morphology.

## Karyotype analysis

After standard cytogenetic harvesting and GTG banding, at least 35 metaphase cells were analyzed and described according to ISCN 2016 (*McGowan-Jordan et al., 2016*).

## Virus production and infection

To create an inducible Cas9 RPTEC cell line, lentiviral production and transduction took place according to the Lenti-X Tet-On 3G Inducible Expression System (Clontech, Takara Bio, Japan) technical manual. In short, Cas9 cDNA was cloned into the pLVX-Tre3G plasmid where after Tre3G-Cas9 and Tet3G lentiviral particles were produced in HEK293T cells. For transduction RPTEC cells were seeded (250 k/well) in a 6-wells plate one day prior to infection. The next day growth media was removed and 1 ml media containing viruses was added. Cells were incubated overnight and after 24 hr media was replaced with 2 ml fresh media. The next day cells were transferred to 10 cm plates and Puromycin (3 µg/ml, Sigma-Aldrich, St. Louis, Missouri, USA) was added to select for successfully transduced cells. G418 was already present in growth media to maintain selective pressure for immortalization and therefore not added for Tet3G selection. For FLCN rescue experiments FLCN cDNA was cloned into pLenti CMVie-IRES-BlastR (gift from Ghassan Mouneimne, Addgene plasmid #119863 [*Puleo et al., 2019*]). The SFPQ-TFE3 fusion sequence was derived from gene expression analysis of a patient-derived pediatric RCC (*Calandrini et al., 2020*) and subsequently cloned as geneblock into pLKO-Ubc lentiviral backbone (*Fumagalli et al., 2017*) using Gibson assembly. For both lentiviral particles were produced in HEK293T and transduced into RPTEC. Blasticidin (15 µg/ml, Invitrogen, Life Technologies) was added for selection of successfully transduced cells and protein overexpression was confirmed by western blotting.

## CRISPR/Cas9 gene editing

For gRNA transfections, RPTEC tet-on Cas9 cells were seeded (75 k cells/well) in 24-well plates with Doxycycline [10 ng/ml, Sigma-Aldrich] at day 0 to induce Cas9 expression. The next day transfection reagent RNAiMAX (Lipofectamine RNAiMAX Transfection Reagent, Thermo Fischer Scientific) was diluted in serum-free medium (Optimem, Gibco, Life Technologies), mixed with gRNA complex (10 uM crRNA and 10 uM tracrRNA, Dharmacon, Horizon Discovery, Cambridge, United Kingdom) and dropwise added to the cells. gRNAs were designed using crispr.mit.edu design tool. Following crRNA sequences were used: FLCN_exon 5 (GTGGCTGACGTATTTAATGG) FLCN_exon 7 (TG TCAGCGATGTCAGCGAGC), TP53_exon 4 (CCATTGTTCAATATCGTCCG), FNIP1_exon 2 (GATA TACAATCAGTCGAATC), and FNIP2_exon 3 (GATGGTTGTACCTGGTACTT).

After 24 h cells were transferred to 10 cm plates and Nutlin-3 (10 µM, Selleck Chemicals, Houston, Texas, USA) was added for selection of TP53 knock-out and thus successfully transfected cells. After selection cells were grown in limiting dilution in 96-wells plates to generate single cell clones. Subsequently, knockout status was assessed by western blot and Sanger sequencing. FLCN^KO RPTEC cell line described in *Figure 8* was created using Synthego's Synthetic cr:tracrRNA Kit and corresponding manual. Cas9/gRNA (FLCN_exon 4 GAGAGCCACGAUGGCAUUCA + modified EZ scaffold) RNP complexes were transfected transiently using Neon Electroporation System (Thermo-Fisher). Subsequently cells were grown in limiting dilution in 96-well plates to generate single-cell clones and knockout status was assessed by western blot and Sanger sequencing. Sequenced samples were analyzed by manual alignment or using the Synthego ICE analysis (ice.synthego.com) tool which gives a quantitative spectrum of *indels* that are formed around the cut site.

## siRNA-mediated knock down

For knock down synthetic siRNAs (siSTAT1_SMARTpool: L-003543-00-0005, siSTAT2_SMARTpool: L-012064-00-0005, siTFEB_SMARTpool: L-009798-00-0005, siTFE3_SMARTpool: L-009363-00-0005, siNT_pool 4: D-001210-04-05, Dharmacon) were transfected using RNAiMAX (Lipofectamine RNAiMAX Transfection Reagent, Thermo Fischer Scientific). After 72 hr 10 nM treatment cells were harvested and stored as dry cell pellet (~1.5E6 cells) in −20°C.

## DNA isolation, PCR, and sequencing

DNA was extracted from dry cell pellet (~1.5E6 cells) according to technical manual of DNA isolation kit (QIAamp DNA Blood Mini Kit, Qiagen, Venlo, Netherlands). Subsequently equal amounts of DNA were amplified by PCR. Tubes were placed in a thermal cycler (Veriti, Thermo-Fischer Scientific) for amplification with specific PCR primer mixes (10 μM). PCR program used for amplification was 1 cycle of 94℃ for 3 min, 5 cycles of 94℃ for 30 s, 65℃ for 30 s, 72℃ for 120 s, 30 cycles of 94℃ for 30 s, 60℃ for 30 s, 72℃ for 2 min, 72℃ for 10 min and ending in a rapid thermal ramp to 10℃. Here after PCR purification (ExoSAP-IT PCR product cleanup (USB products, Affymetrix, Santa Clara, California, USA) and Sephadex G-50 superfine gel (GE Healthcare, Chicago, Illinois, USA)) took place and samples were further analyzed by sequencing. Sequencing was either performed in-house or at Eurofins Genomics. For PCR and sequencing, following primers were used:

```
FLCN_4
Fw 5' GTAAAACGACGGCCAGGGAGGTTTCATGGAGTCAATAGG 3'
Rev 5' CAGGAAACAGCTATGACACTGCTCTCAGGTCCTCC 3'
FLCN_5
Fw 5' GTAAAACGACGGCCAGACCTAAGAGAGTTTGTCGCCCTG 3'
Rev 5' CAGGAAACAGCTATGAAGTGCCTGCCTCCCTGTGC 3'
FLCN_7
Fw 5' GTAAAACGACGGCCAGGGTCCGAGCTGCTGGCAG 3'
Rev 5' CAGGAAACAGCTATGACCAATGTATCGTGACTGCTCTATC 3'
TP53
Fw 5' GAGACCTGTGGGAAGCGAAA 3'
Rev 5' GCTGCCCTGGTAGGTTTTCT 3'
FNIP1
Fw 5' GCCTTTACCAGAGTTTGATCCA 3'
Rev 5' TCATTTCCTTCTCCCTCAGC 3'
FNIP2
Fw 5' CAGTAGCAGCAGCAGCATCT 3'
Rev 5' TCTTCAGCATTCTGCCATCCCA 3'
M13
Fw 5' GTAAAACGACGGCCAG 3'
Rev 5' CAGGAAACAGCTATGA 3'
```

In-house sequencing was done according the BigDye Terminator v1.1 Sequencing Kit (Life Technologies) protocol. For each sample ~75 ng DNA and specific sequencing primers (3.3 μM) were used. Tubes were placed in a thermal cycler for amplification with specific PCR primer mixes. PCR program used for amplification was: 1 cycle of 95℃ for 1 min, 30 cycles of 95℃ for 10 s, 55℃ for 5 s, 60℃ for 2 min and ending in a rapid thermal ramp to 10℃.

## RNA extraction, sequencing and qRT-PCR

RNA was extracted from dry cell pellet (~1.5E6 cells) according to the High Pure RNA Isolation Kit (Roche, Penzberg, Germany) manual. For Illumina-based sequencing, samples were prepped using TruSeq Stranded mRNA Library Preparation Kit according to TruSeq Stranded mRNA Sample Preparation Guide. Sequencing was performed on an Illumina HiSeq 4000 (Illumina, San Diego, California, USA) using run mode SR50. Reads were trimmed using sickle-1.33 (*Joshi and Fass, 2011*) and aligned to hg19 using hisat2-2.0.4 (*Kim et al., 2015*). The alignments were assigned to genes and exons using featurecount-1.5.0-p3 (*Liao et al., 2014*) using the gene annotation provided by the iGenomes resource (*Illumina, 2020*). For quantitative RT-PCR we used Biorad iScript cDNA Synthesis Kit and LightCycler 480 FastStart DNA Master SYBR Green I (Roche). Measurements were performed with LightCycler 480 System and corresponding software (Roche). To determine the quantitative gene expression data levels were normalized to the geometric mean of two housekeeping genes. All experiments were at least performed in duplicate with three technical replicates per experiment. Primer sequences used in this study are:

```
GPNMB
Fw 5' CCTCGTGGGCTCAAATATAAC 3' Rev 5' TTTCTGCAGTTCTTCTCATAGAC 3'
RRAGD
Fw 5' CCTGGCTCTCGTTTGCTTTGTCAG 3' Rev 5' GGGGTGGCTCTCTTTTTCTTCTGC 3'
```

FNIP2
Fw 5' GGTCCTTGGAAGTGGAGCTG 3' Rev 5' GTGAGCGGCCAAAGTTCCT 3'
CTSD
Fw 5' ccatTCCCGAGGTGCTCAAGAACTAC 3' Rev 5' GCAAGCGATGTCCAGCAGTTTG
SQSTM1
Fw 5' ATCGGAGGATCCGAGTGT 3' Rev 5' TGGCTGTGAGCTGCTCTT 3'
WIPI1
Fw 5' GTTGAAGACCCTCCTGGATATTCCTGC 3' Rev 5' gCAGACTGTTTTCAGGGAGTTTCCA
TC 3'
STAT1
Fw 5' CTACGAACATGACCCTATCAC 3' Rev 5' GCTGTCTTTCCACCACAA 3'
ISG15
Fw 5' GAGAGGCAGCGAACTCATCT 3' Rev 5' CTTCAGCTCTGACACCGACA 3'
IFIT1
Fw 5' AGGATGAAGGACAGGAAG 3' Rev 5' GCAGTAAGACAGAAGTGG 3'
MX1
Fw 5' GACAATCAGCCTGGTGGTGGTC 3' Rev 5' GTAACCCTTCTTCAGGTGGAACACG 3'
IRF9
Fw 5' GGGAGCAGTCCATTCAGACA 3' Rev 5' CAGCAGTGAGTAGTCTGGCT 3'
STAT2
Fw 5' CCTCCTGCCTGTGGACATTCG 3' Rev 5' CAGCAACAAGGACTCTGGGTC 3'
OAS2
Fw 5' CAACCTGGATAATGAGTTACCTGC 3' Rev 5' CTGTTGATTGTCGGAAGCAGTTTTC 3'
HPRT1
Fw 5' TGACACTGGGAAAACAATGCA 3' Rev 5 'GGTCCTTTTCACCAGCAAGCT 3'
TBP
Fw 5 'TGCACAGGAGCCAAGAGTGAA 3' Rev 5' CACATCACAGCTCCCCACCA 3'
FLCN
Fw 5 'GGAGAAGCTCGCTGATTTAGAAGAGGA 3' Rev 5' ACCCAGGACCTGCCTCATG 3'
RRAGC
Fw 5 'GGTCTGCATTCTAAGGGAAGAA 3' Rev 5' GAAGTCACACCCACCTCAAA 3'
AMDHD2
Fw 5'TGCTCTCAAGGCACCAAG 3' Rev 5' TGCGTCAGCACCAAAGT 3'
GABARAP
Fw 5 'GGCGAGAAGATCCGAAAGAA 3' Rev 5' GATCAGAAGGCACCAGGTATT 3'
ARHGAP12
Fw 5 'GATACCGGATTCACCAGGAATAG 3' Rev 5' GGGCGTCGTGTAAGAAACT 3'
TFEB
Fw 5 'GCCTGGAGATGACCAACAA 3' Rev 5' CCAGCTCAGCCATGTTCA 3'
TFE3
Fw 5 'AACGACAGGATCAAGGAACTG 3' Rev 5' CGGCTCTCCAGGTCTTTG 3'

Both ISG15 and IFIT1 primers sequences were derived from previous studies (*Bektas et al., 2008*; *Labbé et al., 2012*).

## Differential expression analysis of RNAseq data

We used the R package edgeR (*Robinson et al., 2010*) to compare RNA-sequencing profiles between FLCN^POS and FLCN^NEG replicates, as well as between TP53^POS and TP53^NEG. This involved reading in the gene-level counts, computing library-size normalizing factors using the trimmed-mean of M-values (TMM) method and then fitting a model to estimate the group effect. Obtained p-values were corrected for multiple testing using the Benjamini-Hochberg false discovery rate (FDR) step-up procedure (*Benjamini and Hochberg, 1995*).

## Mass-spectrometry-based proteomics using GeLC-MS/MS

We applied our label-free GeLC-MS/MS-based proteomics workflow with alternating study design that has been extensively bench-marked for reproducibility (*Fratantoni et al., 2010*; *Piersma et al., 2010*; *Piersma et al., 2013*).

## Sample preparation for LC-MS/MS

Equal protein lysates, of each cell line in duplicate, were separated on precast 4–12% gradient gels using NuPAGE SDS-PAGE (Invitrogen, Carlsbad, California, USA). Gels were fixed in 50% ethanol/ 3% phosphoric acid solution and stained with Coomassie R-250. Gel lanes were cut into five bands (see cutting scheme *Figure 2—figure supplement 2B*) and each band was cut into ~1 mm$^3$ cubes. Gel cubes were washed with 50 mM ammonium bicarbonate/50% acetonitrile and were transferred to a microcentrifuge tube, vortexed in 50 mM ammonium bicarbonate for 10 min and pelleted. The supernatant was removed, and the gel cubes were again vortexed in 50 mM ammonium bicarbonate/50% acetonitrile for 10 min. After pelleting and removal of the supernatant, this wash step was repeated. Subsequently, gel cubes were reduced in 50 mM ammonium bicarbonate supplemented with 10 mM DTT at 56°C for 1 hr, where after supernatant was removed. Gel cubes were alkylated in 50 mM ammonium bicarbonate supplemented with 50 mM iodoacetamide for 45 min at RT in the dark. Next, gel cubes were washed with 50 mM ammonium bicarbonate/50% acetonitrile, dried in a vacuum centrifuge at 50°C and covered with trypsin solution (6.25 ng/µl in 50 mM ammonium bicarbonate). Following rehydration with trypsin solution and removal of excess trypsin, gel cubes were covered with 50 mM ammonium bicarbonate and incubated overnight at 25°C. Peptides were extracted from the gel cubes with 100 µl of 1% formic acid (once) and 100 µl of 5% formic acid/50% acetonitrile (twice). A total of 300 µl extracts were stored at −20°C until use. Prior to LC-MS, the extracts were concentrated in a vacuum centrifuge at 50°C, volumes were adjusted to 50 µl by adding 0.05% formic acid, filtered through a 0.45 µm spin filter, and transferred to a LC auto sampler vial.

## LC-MS/MS

Peptides were separated by an Ultimate 3000 nanoLC-MS/MS system (Dionex LC-Packings, Amsterdam, Netherlands) equipped with a 45 cm × 75 µm ID fused silica column custom packed with 1.9 µm 120 Å ReproSil Pur C18 aqua (Dr Maisch GMBH, Ammerbuch-Entringen, Germany). After injection, peptides were trapped at 6 µl/min on a 10 mm ×100 µm ID trap column packed with 5 µm 120 Å ReproSil Pur C18 aqua in 0.05% formic acid. Peptides were separated at 300 nl/min in a 10–40% gradient (buffer A: 0.5% acetic acid (Fisher Scientific), buffer B: 80% ACN, 0.5% acetic acid) in 60 min (100-min inject-to-inject). Eluting peptides were ionized at a potential of +2 kVa into a Q Exactive mass spectrometer (Thermo Fisher, Bremen, Germany). Intact masses were measured at resolution 70,000 (at m/z 200) in the orbitrap using an AGC target value of 3E6 charges. The top 10 peptide signals (charge-states 2+ and higher) were submitted to MS/MS in the HCD (higher-energy collision) cell (1.6 amu isolation width, 25% normalized collision energy). MS/MS spectra were acquired at resolution 17,500 (at m/z 200) in the orbitrap using an AGC target value of 1E6 charges, a maxIT of 60 ms, and an underfill ratio of 0.1%. Dynamic exclusion was applied with a repeat count of 1 and an exclusion time of 30 s.

## Protein identification and label-free quantitation

MS/MS spectra were searched against the reference proteome FASTA file (42161 entries; swissprot_2017_03_human_canonical_and_isoform). Enzyme specificity was set to trypsin, and up to two missed cleavages were allowed. Cysteine carboxamidomethylation (Cys, +57.021464 Da) was treated as fixed modification and methionine oxidation (Met, +15.994915 Da) and N-terminal acetylation (N-terminal, +42.010565 Da) as variable modifications. Peptide precursor ions were searched with a maximum mass deviation of 4.5 ppm and fragment ions with a maximum mass deviation of 20 ppm. Peptide and protein identifications were filtered at an FDR of 1% using the decoy database strategy. The minimal peptide length was seven amino acids. Proteins that could not be differentiated based on MS/MS spectra alone were grouped into protein groups (default MaxQuant settings). Searches were performed with the label-free quantification option selected. Proteins were quantified by spectral counting, that is, the number of identified MS/MS spectra for a given protein (*Liu et al., 2004*) combining the five fractions per sample. Raw counts were normalized on the sum of spectral counts for all identified proteins in a particular sample, relative to the average sample sum determined with all samples. To find statistically significant differences in normalized counts between sample groups, we applied the beta-binomial test (*Pham et al., 2010*), which takes into account within-sample and between-sample variation using an alpha level of 0.05. For proteomic analyses of

tissue material, human kidney lysates (obtained commercially NB820-59231 (HK1) and sc-363764 (HK2)) and fresh BHD tumor tissues (BHDT1 and BHDT2, see details below) lysed in NP40 lysis buffer (50 mM Tris-HCl pH 7.4, 150 mM NaCl, 1% NP40) were used and analysed as described above.

## Patient material

BHD T1 and BHD T2 tumor material was obtained with informed consent for research use according to Amsterdam UMC Medical Ethics Committee guidelines. Both tissues are leftover material from surgery isolated by a pathologist and stored at the Amsterdam UMC Birt-Hogg-Dubé biobank filed under number 2019.359.

## Gene ontology and gene set enrichment analysis

Protein-protein relations were retrieved from the STRING database v11 (*Szklarczyk et al., 2019*) and imported as a network in Cytoscape v3.5.1 (*Shannon et al., 2003*). Enriched gene ontology terms (biological processes) were obtained with the BiNGO v3.0.3 app for Cytoscape (*Maere et al., 2005*), using hypergeometric testing with Benjamini-Hochberg multiple-test correction. Ontology analysis was performed with both the major connected component of the network and subclusters obtained with the ClusterONE v1.0 app for Cytoscape (*Nepusz et al., 2012*). Gene set enrichment was performed with GSEA software (*Subramanian et al., 2005*) using gene sets from Molecular Signatures Database v6.1 (*Liberzon et al., 2011*). Pre-ranked GSEA was performed with ranking by the statistics of the differential FLCN$^{NEG}$-FLCN$^{POS}$ tests. As a ranking metric, we used the product of the sign of the fold change and the negative value of the log10-transformed p-value. Visualization of results was performed in R using the ggplot2 package (*Wickham, 2016*).

## Amino acid starvations and immunofluorescence

For immunofluorescence experiments RPTECs were grown on coverslips (15 mm diameter) on day 1. The medium was refreshed at the end of day 2 or replaced by serum free medium. Amino acid starvation was done with cells that had been serum starved overnight in custom-made DMEM without amino acids but containing 2 mM L-glutamine for 2 hr. Amino acid addback was done by adding an equal volume of starvation medium containing twice the amount of essential amino acids for 15 min. Next, cells were washed twice with ice-cold PBS and fixed in 4% para-formaldehyde for 20 min at room temperature. After washing with PBS, cells were permeabilized with 0.25% Saponin (Sigma Aldrich) in PBS and blocked with blocking solution (10% FCS, 0.25% Saponin in PBS). Coverslips were incubated overnight at room temperature with primary antibodies in blocking solution. The next day, coverslips were washed in 0.25% Saponin in PBS and incubated with secondary antibodies from Fisher Scientific (Alexa 488-goat anti-mouse; 10696113 and Alexa 568-goat anti-rabbit; A11036) for 2 hr at room temperature. Hereafter cells were mounted with Immu-Mount (Thermo Scientific Shandon). Specimens were visualized under Zeiss LSM510 microscope and imaged using Zeiss vision software. The following antibodies were used: mTOR (7C10, CST 2983; 1:300), Lamp2 (H4B4, Ab24631; Abcam; 1:400) and TFE3 (CST 14779; 1:300). For western blot analyses of starvation experiments, RPTECs were seeded in 6-wells plates and treated in like manner. Instead of fixation cells were scraped into 1x Laemmli sample buffer (Sigma Aldrich) and immunoblotted as described below. Starvations, as well as immunofluorescent stainings and immunoblotting were performed three times.

## Immunoblotting

For immunoblotting dry cell pellets (~1.5E6 cells) were lysed in NP40 lysis buffer (50 mM Tris-HCl pH 7.4, 150 mM NaCl, 1% NP40) supplemented with protease and phosphatase inhibitors (Roche). Subsequently samples were boiled at 70°C for 5 min in 1x NuPAGE LDS sample buffer (Novex NP0007, Thermo-Fischer) with 10% 1M DTT (Sigma) and equal amounts were separated by 4–15% or 8–16% SDS-PAGE (BioRad) and blotted onto polyvinylidene fluoride (PVDF) transfer membranes (Merck). After transfer, membrane was blocked for 1 hr at room temperature with 5% milk (ELK, Campina, Amersfoort, Netherlands) in TBST. The primary antibody incubation was ON at 4°C in 2.5% milk in TBST. Next day, membrane was washed and incubated with appropriate secondary antibody (Dako) for 3 hr at 4°C in 2.5% milk in TBST. For detection of phosphorylated proteins, blocking and incubation steps were performed with Bovine Serum Albumin (BSA Fraction V, Roche) instead of milk. After

incubation the membrane was thoroughly washed where after bands were visualized by chemoluminescence (ECL prime, Amersham, VWR, Radnor, Pennsylvania, USA) in combination with ChemiDoc Imaging Systems (BioRad, Hercules, California, USA). For subcellular fractionation experiments the Subcellular Protein Fractionation Kit for Cultured Cells (78840, Thermo Scientific) was used according to manufacturer's protocol.

## Antibodies

For western blot experiments following antibodies were used according to individual datasheets: Vinculin (H-10, sc-25336), FLCN (D14G9, CST 3697S), Cas9 (epigentek A-9000–050), AQP1 (sc-25287), GPNMB (AF2550-SP), SQSTM1 (CST 88588), RRAGD (CST_ 4470S), FNIP1 (ab134969) FNIP2 (HPA042779), STAT2 (GTX103117), pSTAT1 Y701(7649S), TFE3 (HPA023881), H3 (9715S), Tubulin (B-5-1-2, SC-23948), p70S6Kinase T389 (CST 9205), pAKT S473 D9E (CST 4060), total p70S6K (49D7 CST 2708), panAKT 40D4 (CST 2920), 4E-BP1 53H11 (CST 9644), GAPDH (sc-47724) and (MAB374; Merck Millipore).

## Growth curve modeling

We modeled the growth curve data with a random intercept model (*Mirman, 2016*; *Molenberghs and Verbeke, 2000*):

$$Y_{ijk} = (\beta_0 + b_0) + \beta_1 \times \text{Time}_k + \beta_2 \times \text{Time}_k^2 + \beta_3 \times \text{I}_{\text{TP53},i} + \beta_4 \times (\text{Time}_k \times \text{I}_{\text{TP53},i}) + \varepsilon_{ijk}$$

with $Y_{ijk}$ the number of cells of knock-out $j$ at time $k$, $\text{I}_{\text{TP53},j}$ a binary variable indicating if the $i^{th}$ observation is coming from the Cas9-TP53$^{KO}$ ($\text{I}_{\text{TP53},j} = 1$) or from the Cas9-FLCN$^{KO}$ ($\text{I}_{\text{TP53},j} = 0$), $b_0 \sim N(0, \sigma_b^2)$, and $\varepsilon_{ijk} \sim N(0, \sigma_\varepsilon^2)$. *Scheme 1* shows the modeled growth curves for both knock-outs.

Based on the estimated values of the random intercept model (*Table 1*), we can conclude that there is a statistically significant difference ($\alpha$ = 0.05) in the growth curves of Cas9-FLCN$^{KO}$ and Cas9-TP53$^{KO}$ ($\beta_4 \neq 0$; p-value<<0.05), and that there is a statistically significant increase in the number of cells over time ($\beta_1 \neq 0$; p-value<<0.05; and $\beta_2 \neq 0$; p-value<<0.05).

## Scheme 1: Growth rates of FLCNKO and TP53KO cells with fitted curves

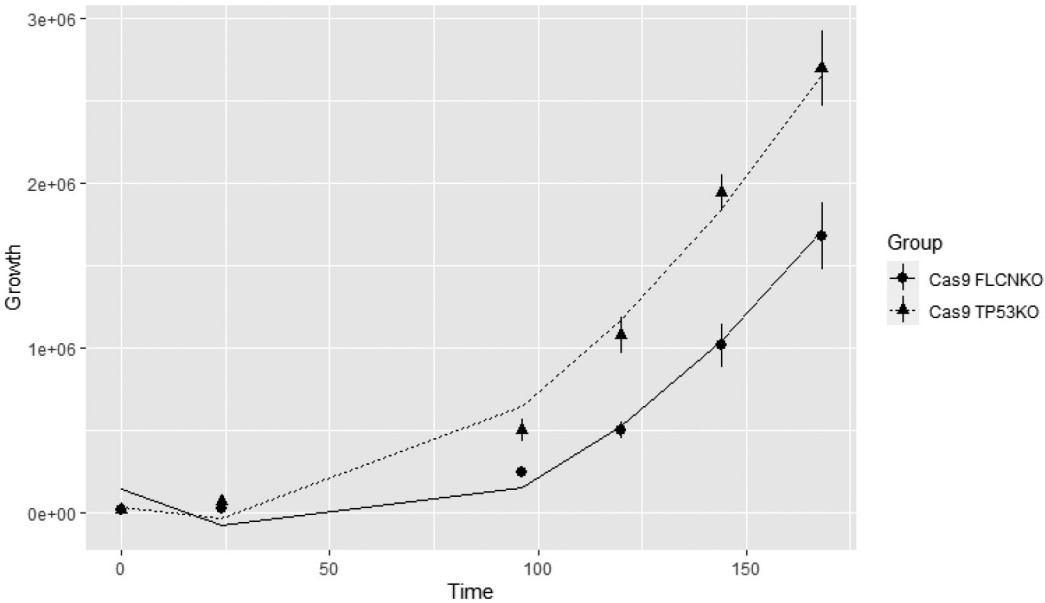

**Scheme 1.** Growth curves of FLCNKO and TP53KO with the fitted curves.

**Table 1.** Estimated values and p-value of the random intercept model.

| | Estimate | Standard error | p-Value |
|---|---|---|---|
| $\beta_0$ (FLCN$^{KO}$ at time 0) | $1.414 \times 10^5$ | $1.079 \times 10^5$ | 0.215 |
| $\beta_1$ (Time) | $-1.222 \times 10^4$ | $1.966 \times 10^3$ | $5.07 \times 10^{-10}$ |
| $\beta_2$ (Time$^2$) | $1.284 \times 10^2$ | $1.138 \times 10^1$ | $<2 \times 10^{-16}$ |
| $\beta_3$ (TP53$^{KO}$) | $-1.080 \times 10^5$ | $1.462 \times 10^5$ | 0.477 |
| (interaction between Time and TP53$^{KO}$) | $6.262 \times 10^3$ | $9.641 \times 10^2$ | $8.31 \times 10^{-11}$ |
| $\sigma_b^2$ | $1.506 \times 10^{10}$ | | |
| $\sigma_\varepsilon^2$ | $6.238 \times 10^{10}$ | | |

## Cytometric bead array and enzyme-linked immunosorbent assay (ELISA)

To detect IFN$\gamma$ in supernatants, the Human IFN-$\gamma$ Flex Set (560111, BD Biosciences, Franklin Lakes, New Jersey, USA) was used according to the manual for the BD CBA Human Soluble Protein Master Buffer Kit. Measurements were done on BD LSRFortessa Flow Cytometer (BD Biosciences) and FCAP Array Software. Supernatants were measured twice in duplicate. For detection of IFN$\alpha$ in supernatants, the VeriKine-HS Human IFN-$\alpha$ All Subtype ELISA kit (#41115, PBL assay science, Piscataway, New Jersey, USA) was used according to manufacturer's protocol. For calculation of IFN$\alpha$ concentrations, blank optical densities were subtracted from standard and sample optical densities. Supernatants were measured twice in duplicate.

## Clonogenic assays

To assess clonogenicity a Crystal Violet (CV) staining was performed. Cell lines were plated in three technical replicates in a six-well plate at low concentration (3000 cells/well). Plates were incubated in a humidified atmosphere at 37°C and 5% $CO_2$ for approximately 10 days until plates displayed colonies with substantially good size. At that moment, cells were fixed with ice-cold methanol (Methanol $\geq$99.8%, Sigma-Aldrich) and plates were incubated for 5 min at RT after which cells were washed with PBS and stained with CV (0.05% solution, Merck) for 30 min at RT. Then CV was washed away with tap water, plates were air dried and scanned for further analysis. Detail photographs ($\times$20) were obtained with AxioScope microscope camera and corresponding software (AxioVision SE64 Rel. 4.9.1, Carl Zeiss, Jena, Germany). At least, two independent experiments with three technical replicates were used to determine colony formation capacities.

## Acknowledgements

We thank the Core Facility Genomics and in particular Daoud Sie for RNA seq and data analysis; we thank Tanja de Gruijl and Ferenc Scheeren for advice and critically reading the manuscript, and Maurice van Steensel and Jeroen van Moorselaar for fruitful discussions. We thank Irene Bijnsdorp for help with mass spec of tissue material and Jürgen Claesen for assistance with statistical analyses. The authors declare that they have no conflicts of interest with the contents of this article.

## Additional information

### Funding

| Funder | Grant reference number | Author |
|---|---|---|
| KWF Kankerbestrijding | Alpe d'Huzes Bas Mulder Award | Jarno Drost |
| Stichting Kinderen Kankervrij | Core Funding | Sepide Derakhshan |
| Oncode Institute | | Jarno Drost |
| Cancer Center Amsterdam | CCA2018-5-51 | Iris E Glykofridis Rob MF Wolthuis |

| Cancer Center Amsterdam | Core Funding Mass Spectrometry Infrastructure | Jaco C Knol<br>Thang V Pham<br>Sander R Piersma<br>Connie R Jimenez |

The funders had no role in study design, data collection and interpretation, or the decision to submit the work for publication.

## Author contributions
Iris E Glykofridis, Conceptualization, Resources, Data curation, Formal analysis, Validation, Investigation, Visualization, Methodology, Writing - original draft, Project administration, Writing - review and editing; Jaco C Knol, Resources, Data curation, Formal analysis, Visualization, Writing-review and editing; Jesper A Balk, Resources, Validation, Investigation; Denise Westland, Sinéad M Lougheed, Validation, Investigation; Thang V Pham, Resources, Data curation, Formal analysis; Sander R Piersma, Resources, Data curation; Sepide Derakhshan, Resources; Puck Veen, Martin A Rooimans, Saskia E van Mil, Pino J Poddighe, Investigation; Franziska Böttger, Visualization, Methodology; Irma van de Beek, Conceptualization, Methodology, Writing - review and editing; Jarno Drost, Resources, Supervision; Fried JT Zwartkruis, Conceptualization, Supervision, Investigation, Visualization, Writing - review and editing; Renee X de Menezes, Resources, Formal analysis, Methodology; Hanne EJ Meijers-Heijboer, Conceptualization, Supervision, Funding acquisition, Writing - review and editing; Arjan C Houweling, Connie R Jimenez, Conceptualization, Supervision, Funding acquisition, Methodology, Writing - review and editing; Rob MF Wolthuis, Conceptualization, Supervision, Funding acquisition, Methodology, Writing - original draft, Project administration, Writing - review and editing

## Author ORCIDs
Iris E Glykofridis (ORCID) https://orcid.org/0000-0003-1829-2403
Rob MF Wolthuis (ORCID) https://orcid.org/0000-0002-3109-1588

## Ethics
Human subjects: BHD T1 and BHD T2 tumor samples were obtained with informed consent. Both tissues are leftover material from surgery and are stored in our BHD biobank (2019.359 at AmsterdamUMC).

## Decision letter and Author response
Decision letter https://doi.org/10.7554/eLife.61630.sa1
Author response https://doi.org/10.7554/eLife.61630.sa2

# Additional files
## Supplementary files
• Supplementary file 1. Table showing differential expression analyses of FLCN$^{POS}$ versus FLCN$^{NEG}$ RPTECs on RNA and protein level.

• Supplementary file 2. Table showing subset of TFE3 targets upregulated in FLCN$^{NEG}$ RPTEC (cluster 3, boxed yellow in *Figure 3A*).

• Transparent reporting form

## Data availability
Data files of transcriptomic and proteomic data are provided as Supplementary file 1. Raw data files deposited on Dryad Digital Repository (RNAseq): https://doi.org/10.5061/dryad.6djh9w0zs and Mass Spec Data are deposited on ProteomeXchange (Mass Spec) under accession number PXD021346.

The following datasets were generated:

| Author(s) | Year | Dataset title | Dataset URL | Database and Identifier |
|---|---|---|---|---|
| Glykofridis IE, Wolthuis RM | 2021 | RNAseq raw counts_FLCN positive vs. FLCN negative RPTECs | http://dx.doi.org/10.5061/dryad.6djh9w0zs | Dryad Digital Repository, 10.5061/dryad.6djh9w0zs |
| Piersma SR, Jimenez CR | 2021 | Loss of FLCN-FNIP1/2 Induces a Non-Canonical Interferon 1 Response in Human Renal Tubular Epithelial Cells | https://www.ebi.ac.uk/pride/archive/projects/PXD021346 | PRIDE, PXD021346 |

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

# Appendix 1

**Appendix 1—key resources table**

| Reagent type (species) or resource | Designation | Source or reference | Identifiers | Additional information |
|---|---|---|---|---|
| Gene (*Homo sapiens*) | FLCN | HUGO Gene Nomenclature Committee | HGNC:27310 | |
| Gene (*Homo sapiens*) | FNIP1 | HUGO Gene Nomenclature Committee | HGNC:29418 | |
| Gene (*Homo sapiens*) | FNIP2 | HUGO Gene Nomenclature Committee | HGNC:29280 | |
| Gene (*Homo sapiens*) | TFE3 | HUGO Gene Nomenclature Committee | HGNC:11752 | |
| Gene (*Homo sapiens*) | TFEB | HUGO Gene Nomenclature Committee | HGNC:11753 | |
| Gene (*Homo sapiens*) | STAT1 | HUGO Gene Nomenclature Committee | HGNC:11362 | |
| Gene (*Homo sapiens*) | STAT2 | HUGO Gene Nomenclature Committee | HGNC:11363 | |
| Cell line (*Homo sapiens*) | RPE-1 tet on Cas9 TP53$^{KO}$ | *Benedict et al., 2020* | PMID:32084359 | Originally derived from hTERT RPE-1 (ATCC Cat# CRL-4000, RRID:CVCL_4388) |
| Cell line (*Homo sapiens*) | RPE tet on Cas9 TP53$^{KO}$ FLCN$^{KO}$ C2 | This paper | | knock out cell lines, see Material and methods section CRISPR/Cas9 gene editing |
| Cell line (*Homo sapiens*) | RPTEC/TERT1 | ATCC | ATCC Cat# CRL-4031, RRID:CVCL_K278 | |
| Cell line (*Homo sapiens*) | - RPTEC tet on Cas9<br>- RPTEC tet on Cas9 TP53$^{KO}$ (pool and three clones)<br>- RPTEC tet on Cas9 TP53$^{KO}$ FLCN$^{KO}$ C1-3<br>- RPTEC RPTEC tet on Cas9 TP53$^{wt}$ FLCN$^{KO}$<br>- RPTEC tet on Cas9 FNIP1/FNIP2$^{KO}$<br>- RPTEC FLCN$^{KO}$ | This paper | | knock out cell lines, see Material and methods section CRISPR/Cas9 gene editing |
| Cell line (*Homo sapiens*) | - RPTEC SFPQ-TFE3<br>- RPTEC SFPQ-TFE3 FLCN$^{KO}$ | This paper | | Lentivirally transduced SFPQ-TFE3 mutant, with and without CRISPR mediated FLCN knock out |
| Sequenced-based reagent (human) | siRNA STAT1 | Dharmacon, Horizon discovery | L-003543-00-0005 | siRNA pool used for gene knock down experiments |

*Continued on next page*

*Appendix 1—key resources table continued*

| Reagent type (species) or resource | Designation | Source or reference | Identifiers | Additional information |
|---|---|---|---|---|
| Sequenced-based reagent (human) | siRNA STAT2 | Dharmacon, Horizon discovery | L-012064-00-0005 | siRNA pool used for gene knock down experiments |
| Sequenced-based reagent (human) | siRNA TFEB | Dharmacon, Horizon discovery | L-009798-00-0005 | siRNA pool used for gene knock down experiments |
| Sequenced-based reagent (human) | siRNA TFE3 | Dharmacon, Horizon discovery | L-009363-00-0005 | siRNA pool used for gene knock down experiments |
| Sequenced-based reagent (human) | siRNA non-targeting control | Dharmacon, Horizon discovery | D-001210-04-05 | siRNA pool used for gene knock down experiments |
| Transfected construct (human) | pLenti CMVie-IRES-BlastR FLCN cDNA | This paper | | FLCN rescue by overexpression of cDNA in Addgene plasmid #119863 (*Puleo et al., 2019*) |
| Transfected construct (human) | pLKO-Ubc SFPQ-TFE3 | *Fumagalli et al., 2017* | PMID:28270604 | Patient derived SFPQ-TFE3 fusion sequence transduced in RPTEC |
| Sequenced-based reagent (*Homo sapiens*) | crRNA FLCN_exon 5 (GTGGC TGACGTATTTAA TGG) | Dharmacon, Horizon Discovery | | Synthetic gRNA for CRISPR/Cas9 mediated gene knock out |
| Sequenced-based reagent (*Homo sapiens*) | crRNA FLCN_exon 7 (TGTCAGCGA TGTCAGCGAGC) | Dharmacon, Horizon Discovery | | Synthetic gRNA for CRISPR/Cas9 mediated gene knock out |
| Sequenced-based reagent (*Homo sapiens*) | crRNATP53_exon 4 (CCATTGTTCAA TATCGTCCG) | Dharmacon, Horizon Discovery | | Synthetic gRNA for CRISPR/Cas9 mediated gene knock out |
| Sequenced-based reagent (*Homo sapiens*) | crRNA FNIP1_exon 2 (GATATACAA TCAGTCGAATC) | Dharmacon, Horizon Discovery | | Synthetic gRNA for CRISPR/Cas9 mediated gene knock out |
| Sequenced-based reagent (*Homo sapiens*) | crRNA FNIP2_exon 3 (GATGGTTG TACCTGGTAC TT) | Dharmacon, Horizon Discovery | | Synthetic gRNA for CRISPR/Cas9 mediated gene knock out |
| Sequenced-based reagent (*Homo sapiens*) | FLCN_exon 4 GAGAGCCACGA UGGCAUUCA + modified EZ scaffold | Synthego | | Synthetic gRNA for CRISPR/Cas9 mediated gene knock out |
| Biological sample (*Homo sapiens*) | BHD kidney tumor 1 | This paper | | BHD T1 sample for mass spectrometry, see Material and methods section Patient material |
| Biological sample (*Homo sapiens*) | BHD kidney tumor 2 | This paper | | BHD T2 sample for mass spectrometry, see Material and methods section Patient material |
| Biological sample (*Homo sapiens*) | Human kidney lysate 1 | Novus Bio | NB820-59231 | HK1 sample for mass spectrometry |
| Biological sample (*Homo sapiens*) | Human kidney lysate 2 | Santa Cruz | sc-363764 | HK2 sample for mass spectrometry |

*Continued on next page*

*Appendix 1—key resources table continued*

| Reagent type (species) or resource | Designation | Source or reference | Identifiers | Additional information |
|---|---|---|---|---|
| Antibodies (for westerns) | Vinculin (mouse mAb, H-10) | Santa Cruz | sc-25336 | (1:1000) |
| | FLCN (rabbit mAb, D14G9) | Cell Signalling | CST 3697S | (1:1000) |
| | Cas9 (mouse mAb, 7A9) | Epigentek | A-9000–050 | (1:1000) |
| | AQP1 (mouse mAb, B11) | Santa Cruz | sc-25287 | (1:100) |
| | GPNMB (goat pAb) | R and D systems | AF2550-SP | (0.5 µg/mL) |
| | SQSTM1 (mouse mAb, D5L7G) | Cell Signalling | CST 88588 | (1:1000) |
| | RRAGD (rabbit pAb) | Cell Signalling | CST 4470S | (1:1000) |
| | FNIP1 (rabbit mAb) | Abcam | ab134969 | (1:1000) |
| | FNIP2 (rabbit pAb) | Atlas Antibodies | HPA042779 | (1:1000) |
| | STAT2 (rabbit pAb) | GeneTex | GTX103117 | (1:1000) |
| | pSTAT1 Y701 (rabbit mAb, D4A7) | Cell Signalling | CST 7649S | (1:1000) |
| | TFE3 (rabbit pAb) | Atlas Antibodies | HPA023881 | (1:1000) |
| | H3 (rabbit pAb) | Cell Signalling | CST 9715S | (1:1000) |
| | αTubulin (mouse mAb, B-5-1-2) | Santa Cruz | sc-23948 | (1:2000) |
| | p70S6Kinase T389 (rabbit pAb) | Cell Signalling | CST 9205 | (1:1000) |
| | pAKT S473 (rabbit mAb, D9E) | Cell Signalling | CST 4060 | (1:2000) |
| | total p70S6K (rabbit mAb, 49D7) | Cell Signalling | CST 2708 | (1:1000) |
| | panAKT (mouse mAb 40D4) | Cell Signalling | CST 2920 | (1:2000) |
| | 4E-BP1 (rabbit mAb 53H11) | Cell Signalling | CST 9644 | (1:1000) |
| | GAPDH (mouse mAb, 0411) | Santa Cruz | sc-47724 | (1:5000) |
| | GAPDH (mouse mAb, 6C5) | Merck Millipore | MAB374 | (1:200) |

*Appendix 1—key resources table continued*

| Reagent type (species) or resource | Designation | Source or reference | Identifiers | Additional information |
|---|---|---|---|---|
| Antibodies (for immunofluorescence) | **mTOR** (rabbit mAb, 7C10), | Cell Signalling | CST 2983 | (1:300) |
| | **Lamp2** (mouse mAb, H4B4) | Abcam | ab25631 | (1:400) |
| | **TFE3** (rabbit pAb) | Cell Signalling | CST 14779 | (1:300) |
| Sequence-based reagent | qRT-PCR and sequencing primers | Sigma-Aldrich | | Described in corresponding material and method sections |
| Commercial assay or kit | Lenti-X Tet-On 3G Inducible Expression System | Clontech, Takara Bio | 631187 | Creation of lentiviral constructs to generate Doxycycline inducible Cas9 cell line |
| Commercial assay or kit | High Pure RNA Isolation Kit | Roche | 11828665001 | RNA isolation kit for RNAseq and qRT-PCR analyses |
| Commercial assay or kit | iScript cDNA Synthesis Kit | Bio-Rad | 170–8891 | cDNA synthesis kit for qRT-PCR analyses |
| Commercial assay or kit | IFN-γ Flex Set CBA | BD Biosciences | 560111 | Flow cytometry based Cytometric bead array |
| Commercial assay or kit | VeriKine-HS Human IFN-α All Subtype ELISA kit | PBL assay science | 41115 | IFN-α Enzyme-Linked Immunosorbent Assay |
| Chemical compound, drug | Crystal Violet | | 1014080025 | Stain clonogeniticy using a (0,05% solution) |
| Software, algorithm | R/Rstudio | | | edgeR (*Robinson et al., 2010*) ggplot (*Wickham, 2016*) |
| Software, algorithm | Cytoscape | *Shannon et al., 2003* | PMID:14597658 | iRegulon BinGO ClusterOne v1.0 (*Nepusz et al., 2012*) |
| Software, algorithm | GSEA MSigDB | *Subramanian et al., 2005* *Liberzon et al., 2011* | PMID:16199517 PMID:26771021 | Gene set enrichment analyses |
| Software, algorithm | GraphPad Prism | | RRID:SCR_002798 | Rel. 8.2.2, plots and graph design |
| Software, algorithm | AxioVision SE64 | Carl Zeiss | Rel. 4.9.1 | Microscope camera software |

