## [Decision Letter]

**Acceptance summary:**

The authors have very elegantly provided more mechanistic insights on the role of Folliculin in renal cancer development, of which germline mutants are known to predispose to this malignancy. In addition, the work pinpoints potential candidate prognostic biomarkers, which may be of translational value for patients.

**Decision letter after peer review:**

Thank you for submitting your article "Loss of FLCN-FNIP1/2 Induces a Non-Canonical Interferon Response in Human Renal Tubular Epithelial Cells" for consideration by *eLife*. Your article has been reviewed by two peer reviewers, and the evaluation has been overseen by a Reviewing Editor and Kevin Struhl as the Senior Editor. The reviewers have opted to remain anonymous.

The reviewers have discussed the reviews with one another and the Reviewing Editor has drafted this decision to help you prepare a revised submission.

Glykofridis and colleagues have investigated the consequences of knocking out FLCN and its partners FNIP1/2 in a human renal proximal tubule cell model through RNA sequencing, proteomics and pathway analysis. They have identified two transcriptional signatures that are induced upon FLCN loss: (1) TFE3 transcriptional activation, which supports other published data, and (2) upregulation of genes that are under the control of ISREs and directed through upregulation of STAT1 and 2 in an interferon-independent manner. This is a novel finding. They demonstrate upregulation of TFE3 target genes and genes regulated by ISREs in 2 BHD tumors. These findings confirm TFE3 as a main target of the FLCN-FNIP1/2 axis and present a novel ISRE gene activation program which is activated by FLCN loss and may represent a new area to explore for biomarkers and targeted therapies.

The following issues would need to be addressed for consideration of this work in *eLife*:

1) The authors state that FLCN-knockout induces a TFE3-dictated genetic program, and while their data do suggest such a connection, they don't formally show this. A key experiment do address this, is to perform a TFE3 knockdown/knockout experiment, followed by RNA-seq (or minimal: an extensive set of QPCRs) in FLCN-knockout cells. This way, the authors can support their claim the genes diff. regulated upon FLCN knockout are indeed under TFE3 control (which they now claim, but don't show). This experiment should be added in a revised version of the manuscript.

2) Throughout the text, the authors should tone down the clinical implications of their work; such strong clinical statements (e.g. at the end of the Introduction) cannot be claimed with the current work, definitively not being an n=1 cell line study.

3) Include in the Discussion section that the study is based on n=1 cell line, and should be treated as such.

---

## [Author Response]

[…] The following issues would need to be addressed for consideration of this work in eLife:1) The authors state that FLCN-knockout induces a TFE3-dictated genetic program, and while their data do suggest such a connection, they don't formally show this. A key experiment do address this, is to perform a TFE3 knockdown/knockout experiment, followed by RNA-seq (or minimal: an extensive set of QPCRs) in FLCN-knockout cells. This way, the authors can support their claim the genes diff. regulated upon FLCN knockout are indeed under TFE3 control (which they now claim, but don't show). This experiment should be added in a revised version of the manuscript.

We thank the reviewer for these constructive comments. Summarizing, we showed that TFE3 resides constitutively in the nucleus in the absence of FLCN, and that FLCN loss causes gene induction of confirmed TFE3/TFEB targets under the control of E-box elements. However, we indeed agree that we had not directly addressed a TFE-dependent activation of these FLCN controlled E-box genes yet. We anticipated that these experiments might be slightly complicated by the expression of different TFE family members with highly overlapping function and target genes. Nevertheless, reassuringly in our expression datasets we observed that TFE3 is indeed the dominantly expressed TFE family member in RPTEC-TERT cells, together with its close family member TFEB, see Figure 6E.

To address the concern of the reviewer, we therefore now included siRNA experiments to knock down TFE3 as well as the combination of TFE3 and TFEB. While we only achieved partial knock down of these TFE members, it was clear that co-depletion of TFE3 and TFEB repressed the induction of GPNMB, RRAGD, RRAGC, AMDHD2, and SQSTM1 (p62) in three independent TP53^KO^FLCN^KO^ RPTEC clones of different karyotypes, while showing modest repression of several other E-box genes.

These data are now shown in a new Figure 3D.

Furthermore, a new Figure 3—figure supplement 1C shows that some of these genes, notably GPNMB and RRAGD which are induced 20-200x by FLCN loss (Figure 3B), as well as the lysosomal marker AMDHD2, are effectively reverted by TFE3 siRNA. The reduced induction of these FLCN target genes strongly correlated with the reduced expression levels of TFE3. These experiments therefore pinpoint to TFE3 and to a lesser extent TFEB as transcription factors activated by FLCN loss in human renal epithelial cells, similar to the situation very recently described in mice (Napolitano et al., 2020).

Finally, we could clearly confirm the rescuing effects of TFE3/TFEB knock down in a more recently established TP53^WT^FLCN^KO^ RPTEC cell line, of which the results are in shown in a new Figure 8—figure supplement 1C. In these cells we achieved a more effective knock-down of TFE3 and TFEB, resulting in effective repression of gene induction of FLCN targets GPNMB, RRAGD, RRAGC, AMDHD2, SQSTM1, GAPARAP, ARHGAP12, WIP1 and FNIP2.

While we also appreciate the suggestion to perform RNAseq experiments to further investigate the detailed consequences of TFE3 and/or TFEB knock-out on the repertoire and dynamics of target expression, we believe that this amounts to a larger study in which also many different conditions and time points would need to be included to account for the effects of feed-back mechanisms and differences in mRNA stabilities. While highly interesting, for the current study we decided to focus on a defined set of genes highly upregulated upon FLCN loss.

Taken together, we feel that we now can claim that FLCN-knockout induces two independent gene expression programs, one TFE3/TFEB-dependent and, as the main novelty of our manuscript, a second ISRE/STAT-dictated program.

In the revised manuscript, we mended the text of the Abstract and Results sections and Discussion to illustrate these findings. Furthermore, we restructured the order of the figures to make room for the new data. Therefore, the previous Figure 3D (western blot) is now moved to a new –mTOR focused – main Figure 4 combined with previously supplementary Figure 4C (immunofluorescence stainings).

2) Throughout the text, the authors should tone down the clinical implications of their work; such strong clinical statements (e.g. at the end of the Introduction) cannot be claimed with the current work, definitively not being an n=1 cell line study.3) Include in the Discussion section that the study is based on n=1 cell line, and should be treated as such.

We appreciate the reviewer’s concerns about the clinical implications of our work and toned down our clinical statements throughout the manuscript, including the phrasing in the Introduction. To address point 2 and 3, we now clearly pointed out that our study is based on derivatives of a single cell line (Discussion). On the other hand, we also wish to mention that: (a) the human RPTEC/TERT1 cell line we used is widely accepted as a physiologically relevant in vitro cell model to study kidney tubular functions, and (b) we included multiple, karyotypically different subclones as well as two independent BHD tumors in our analyses, which we used to cross-validate our results.